# Selective Prediction via Training Dynamics

**Stephan Rabanser**                                    *stephan@cs.toronto.edu*
*University of Toronto & Vector Institute*

**Anvith Thudi**                                    *anvith.thudi@mail.utoronto.ca*
*University of Toronto & Vector Institute*

**Kimia Hamidieh**                                    *hamidieh@mit.edu*
*Massachusetts Institute of Technology*

**Adam Dziedzic**                                    *adam.dziedzic@cispa.de*
*CISPA Helmholtz Center for Information Security*

**Israfil Bahceci**                                    *israfil.bahceci@ericsson.com*
*Ericsson*

**Akram Bin Sediq**                                    *akram.bin.sediq@ericsson.com*
*Ericsson*

**Hamza Sokun**                                    *hamza.sokun@ericsson.com*
*Ericsson*

**Nicolas Papernot**                                    *nicolas.papernot@utoronto.ca*
*University of Toronto & Vector Institute*

**Reviewed on OpenReview:** *https://openreview.net/forum?id=2wgnepQjyF*

## Abstract

Selective Prediction is the task of rejecting inputs a model would predict incorrectly on. This involves a trade-off between input space coverage (how many data points are accepted) and model utility (how good is the performance on accepted data points). Current methods for selective prediction typically impose constraints on either the model architecture or the optimization objective; this inhibits their usage in practice and introduces unknown interactions with pre-existing loss functions. In contrast to prior work, we show that state-of-the-art selective prediction performance can be attained solely from studying the (discretized) training dynamics of a model. We propose a general framework that, given a test input, monitors metrics capturing the instability of predictions from intermediate models (i.e., checkpoints) obtained during training w.r.t. the final model's prediction. In particular, we reject data points exhibiting too much disagreement with the final prediction at late stages in training. The proposed rejection mechanism is domain-agnostic (i.e., it works for both discrete and real-valued prediction) and can be flexibly combined with existing selective prediction approaches as it does not require any train-time modifications. Our experimental evaluation on image classification, regression, and time series problems shows that our method beats past state-of-the-art accuracy/utility trade-offs on typical selective prediction benchmarks.

## 1 Introduction

Machine learning (ML) is increasingly deployed in high-stakes decision-making environments with strong reliability and safety requirements. One of these requirements is the detection of inputs for which the ML model produces an erroneous prediction. This is particularly important when deploying deep neural networks

Figure 1: **Our proposed SPTD method for a classification example**. We store checkpoints of intermediate models during model training. At inference time, given a test input, we compute various metrics capturing the stability of intermediate predictions with respect to the final model prediction. Data points with high stability are accepted, data points with low stability are rejected.

(DNNs) for applications with low tolerances for false-positives (i.e., classifying with a wrong label), such as healthcare (Challen et al., 2019; Mozannar & Sontag, 2020), self-driving (Ghodsi et al., 2021), and law (Vieira et al., 2021). This problem setup is captured by the Selective Prediction (SP) framework, which introduces an accept/reject function (a so-called *gating mechanism*) to abstain from predicting on individual test points in the presence of high prediction uncertainty (Geifman & El-Yaniv, 2017). Specifically, SP aims to (i) only accept inputs on which the ML model would achieve high utility, while (ii) maintaining high coverage (i.e., correctly accepting as many inputs as possible).

Current selective prediction techniques take one of two directions: (i) augmentation of the architecture of the underlying ML model (Geifman & El-Yaniv, 2019); or (ii) training the model using a purposefully adapted loss function (Liu et al., 2019; Huang et al., 2020; Gangrade et al., 2021). The unifying principle behind these methods is to modify the training stage in order to accommodate selective prediction. While many ad-hoc experimentation setups are amenable to these changes, productionalized environments often impose data pipeline constraints which limit the applicability of existing methods. Such constraints include, but are not limited to, data access revocation, high (re)-training costs, or pre-existing architecture/loss modifications whose interplay with selective prediction adaptations are unexplored. As a result of theses limitations, selective prediction approaches are hard to deploy in production environments.

We instead show that *these modifications are unnecessary*. That is, our method, which **establishes new SOTA results for selective prediction** across a variety of datasets, not only outperforms existing work but **our method can be easily applied on top of all existing models**, unlike past methods. Moreover, our method is not restricted to classification problems but can be applied for real-valued prediction problems, too, like regression and time series prediction tasks. This is an important contribution as recent SP approaches have solely focused on improving selective *classification*.

Our approach builds on the following observation: typical DNNs are trained using an iterative optimization procedure, e.g., using Stochastic Gradient Descent (SGD). Due to the sequential nature of this optimization process, as training goes on, the optimization process yields a sequence of intermediate models. Current selective prediction methods rely only on the final model, ignoring valuable statistics available from the model's training sequence. In this work, however, we propose to take advantage of the information contained in these optimization trajectories for the purpose of selective prediction. By studying the usefulness of these trajectories, we observe that instability in SGD convergence is often indicative of high aleatoric uncertainty (i.e., irreducible data noise such as overlap between distinct data classes). Furthermore, past work on example difficulty (Jiang et al., 2020; Toneva et al., 2018; Hooker et al., 2019; Agarwal et al., 2020) has highlighted faster convergence as indicative of easy-to-learn training examples (and conversely slow convergence of hard-to-learn training examples). We hypothesize that such training time correlations with uncertainty also hold for test points and studying how test time predictions evolve over the intermediate checkpoints is useful for reliable uncertainty quantification.

With this hypothesis, we derive the first framework for **S**elective **P**rediction based on neural network **T**raining **D**ynamics (**SPTD**, see Figure 1 for an example using a classification setup). Through a formalization

of this particular neural network training dynamics problem, we first note that a useful property of the intermediate models' predictions for a test point is whether they converge ahead of the final prediction. This convergence can be measured by deriving a prediction instability score measuring how strongly predictions of intermediate models agree with the final model. While the exact specifics of how we measure instability differs between domains (classification vs regression), our resulting score generalizes across application domains and measures weighted prediction instability. This weighting allows us to emphasize instability late in training which we deem indicative of points that should be rejected. Note that this approach is transparent w.r.t. the training stage: our method only requires that intermediate checkpoints were recorded when a model was trained, which is an established practice (especially when operating in shared computing environments such as GPU clusters). Moreover, when compared to competing ensembling-based methods, such as Deep Ensembles (Lakshminarayanan et al., 2017), our approach can match the same inference-time cost while being significantly cheaper to train.

To summarize, our main contributions are as follows:

1. We present a motivating synthetic example using a linear model, showcasing the effectiveness of training dynamics information in the presence of a challenging classification task (Section 3.1).

2. We propose a novel method for selective prediction based on training dynamics (**SPTD**, Section 3.2). To that end, we devise an effective scoring mechanism capturing weighted prediction instability of intermediate models with the final prediction for individual test points. Our methods allow for selective classification, selective regression, and selective time series prediction. Moreover, **SPTD** can be applied to all existing models whose checkpoints were recorded during training.

3. We highlight an in-depth connection between our **SPTD** approach and forging (Thudi et al. (2022), Section 3.3), which has shown that optimizing a model on distinct datasets can lead to the same sequence of checkpoints. This connection demonstrates that our metrics can be motivated from a variety of different perspectives.

4. We perform a comprehensive set of empirical experiments on established selective prediction benchmarks spanning over classification, regression, and time series prediction problems (Section 4). Our results obtained from all instances of **SPTD** demonstrate highly favorable utility/coverage trade-offs, establishing new state-of-the-art results in the field at a fraction of the training time cost of competitive prior approaches.

## 2 Background on Selective Prediction

**Supervised Learning Setup.** Our work considers the standard supervised learning setup. We assume access to a dataset $D = \{(\boldsymbol{x}_i, y_i)\}_{i=1}^{M}$ consisting of $M$ data points $(\boldsymbol{x}, y)$ with $\boldsymbol{x} \in \mathcal{X}$ and $y \in \mathcal{Y}$. We refer to $\mathcal{X} := \mathbb{R}^d$ as the covariate space (or input/data space) of dimensionality $d$. For classification problems, we define $\mathcal{Y} := [C] = \{1, 2, \ldots, C\}$ as the label space consisting of $C$ classes. For regression and time series problems (such as demand forecasting) we instead define $\mathcal{Y} := \mathbb{R}$ and $\mathcal{Y} := \mathbb{R}^R$ respectively (with $R$ being the prediction horizon). All data points $(\boldsymbol{x}, y)$ are sampled independently from the underlying distribution $p$ defined over the joint covariate and label spaces $\mathcal{X} \times \mathcal{Y}$. Our goal is to learn a prediction function $f : \mathcal{X} \to \mathcal{Y}$ which minimizes the risk $\mathcal{R}(f_{\boldsymbol{\theta}})$ with respect to the underlying data distribution $p$ and an appropriately chosen loss function $\ell : \mathcal{Y} \times \mathcal{Y} \to \mathbb{R}$. We can derive the optimal parameters $\hat{\boldsymbol{\theta}}$ via empirical risk minimization which approximates the true risk $\mathcal{R}(f_{\boldsymbol{\theta}})$ through sampling, thereby ensuring that $\boldsymbol{\theta}^* \approx \hat{\boldsymbol{\theta}}$ for a sufficiently large amount of samples:

$$\boldsymbol{\theta}^* = \arg\min_{\boldsymbol{\theta}} \mathcal{R}(f_{\boldsymbol{\theta}}) = \arg\min_{\boldsymbol{\theta}} \mathbb{E}_{p(\boldsymbol{x},y)}[\ell(f_{\boldsymbol{\theta}}(\boldsymbol{x}), y)] \tag{1}$$

$$\hat{\boldsymbol{\theta}} = \arg\min_{\boldsymbol{\theta}} \hat{\mathcal{R}}(f_{\boldsymbol{\theta}}) = \arg\min_{\boldsymbol{\theta}} \frac{1}{M} \sum_{i=1}^{N} \ell(f_{\boldsymbol{\theta}}(\boldsymbol{x}_i), y_i) \tag{2}$$

In the following, we drop the explicit dependence of $f$ on $\boldsymbol{\theta}$ and simply denote the predictive function by $f$.

**Selective Prediction Setup.** Selective prediction alters the standard supervised learning setup by introducing a rejection state $\perp$ through a *gating mechanism* (El-Yaniv & Wiener, 2010). In particular, such a mechanism introduces a selection function $g : \mathcal{X} \to \mathbb{R}$ which determines if a model should predict on a data point $\boldsymbol{x}$. Given an acceptance threshold $\tau$, the resulting predictive model can be summarized as:

$$(f,g)(\boldsymbol{x}) = \begin{cases} f(\boldsymbol{x}) & g(\boldsymbol{x}) \leq \tau \\ \perp & \text{otherwise.} \end{cases} \tag{3}$$

**Selective Prediction Evaluation Metrics.** Prior work evaluates the performance of a selective predictor $(f,g)$ based on two metrics: the *coverage* of $(f,g)$ (i.e., what fraction of points we predict on) and the *selective utility* of $(f,g)$ on the accepted points. Note that the exact utility metric depends on the type of the underlying selective prediction task (e.g. accuracy for classification, $R^2$ for regression, and a quantile-based loss for time series forecasting). Successful SP methods aim to obtain both strong selective utility and high coverage. Note that these two metrics are at odds with each other: naïvely improving utility leads to lower coverage and vice-versa. The complete performance profile of a model can be specified using the risk–coverage curve, which defines the risk as a function of coverage (El-Yaniv & Wiener, 2010). These metrics can be formally defined as follows:

$$\text{coverage}(f,g) = \frac{M_\tau}{M} \tag{4}$$

$$\text{utility}(f,g) = \sum_{\{(\boldsymbol{x},y):g(\boldsymbol{x}) \leq \tau\}} u(f(\boldsymbol{x}), y) \tag{5}$$

Here, $u(\cdot, \cdot)$ corresponds to the specifically used utility function, $M_\tau = \sum_{i=1}^{M} \mathbb{1}[\boldsymbol{x}_i : g(\boldsymbol{x}_i) \leq \tau]$ corresponds to the number of accepted data points at threshold $\tau$, and $\mathbb{1}[\cdot]$ corresponds to the indicator function. We define the following utility functions to be used based on the problem domain:

1. *Classification*: We use accuracy on accepted points as our utility function for classification:

$$\text{Accuracy} = \frac{1}{M_\tau} \sum_{i=1}^{M_\tau} \mathbb{1}[\boldsymbol{x}_i : f(\boldsymbol{x}_i) = y_i] \tag{6}$$

2. *Regression*: We use the coefficient of determination ($R^2$ score, which is a scaled version of the mean squared error) on accepted points as our utility function for regression:

$$R^2 = 1 - \frac{\sum_{i=1}^{M_\tau} (f(\boldsymbol{x}_i) - y_i)^2}{\sum_{i=1}^{M_\tau} (y_i - \frac{1}{M_\tau} \sum_{j=1}^{M_\tau} y_j)^2} \tag{7}$$

3. *Time Series Forecasting*: We use the Mean Scaled Interval Score (MSIS) Gneiting & Raftery (2007) on accepted series as our utility function for time series forecasting

$$\text{MSIS} = \frac{1}{M_\tau R} \sum_{i=1}^{M_\tau} \frac{\sum_{r=n+1}^{n+R} (u_{i,r} - l_{i,r}) + \frac{2}{\alpha} (l_{i,r} - y_{i,r}) \mathbb{1}[y_{i,r} < l_{i,r}] + \frac{2}{\alpha} (y_{i,r} - u_{i,r}) \mathbb{1}[y_{i,r} > u_{i,r}]}{\frac{1}{n-m} \sum_{r=m+1}^{n} |y_{i,r} - y_{i,r-m}|} \tag{8}$$

where $\alpha$ refers to a specific predictive quantile, $n$ to the conditioning length of the time series, $m$ to the length of the seasonal period, and $u_{i,r}$ and $l_{i,r}$ to the upper and lower bounds on the prediction range, respectively.

## 2.1 Past & Related Work

**Softmax Response Baseline (classification).** The first work on selective classification was the softmax response (**SR**) mechanism (Hendrycks & Gimpel, 2016; Geifman & El-Yaniv, 2017). A classification model

typically has a softmax output layer which takes in unnormalized activations in $z_i \in \mathbb{R}^C$ (referred to as logits) from a linear model or a deep neural net. These activations are mapped through the softmax function which normalizes all entries

$$\sigma(\boldsymbol{z})_i = \frac{e^{z_i}}{\sum_{j=1}^{K} e^{z_j}} \tag{9}$$

to the interval $[0,1]$ and further ensures that $\sum_{i=1}^{C} \sigma(\boldsymbol{z})_i = 1$. As a result, the softmax output can be interpreted as a conditional probability distribution which we denote $f(y|\boldsymbol{x})$. The softmax response mechanism applies a threshold $\tau$ to the maximum response of the softmax layer: $\max_{y \in \mathcal{Y}} f(y|\boldsymbol{x})$. Given a confidence parameter $\delta$ and desired risk $\hat{\mathcal{R}}(f)$, **SR** constructs $(f, g)$ with test error no larger than $\hat{\mathcal{R}}(f)$ with probability $\geq 1 - \delta$. While this approach is simple to implement, it has been shown to produce over-confident results due to poor calibration of deep neural networks (Guo et al., 2017).[1]

**Loss Modifications (mostly classification).** The first work to deliberately address selective classification via architecture modification was SelectiveNet (Geifman & El-Yaniv, 2019), which trains a model to jointly optimize for classification and rejection. A loss penalty is added to enforce a particular coverage constraint using a variant of the interior point method Potra & Wright (2000) which is often used for solving linear and non-linear convex optimization problems. To optimize selective accuracy over the full coverage spectrum in a single training run, Deep Gamblers (Liu et al., 2019) transform the original $C$-class problem into a $(C+1)$-class problem where the additional class represents model abstention. A similar approach is given by Self-Adaptive Training (**SAT**) (Huang et al., 2020) which also uses a $(C+1)$-class setup but instead incorporates an exponential average of intermediate predictions into the loss function. Other similar approaches include: performing statistical inference for the marginal prediction-set coverage rates using model ensembles (Feng et al., 2021), confidence prediction using an earlier snapshot of the model (Geifman et al., 2018), estimating the gap between classification regions corresponding to each class (Gangrade et al., 2021), and complete precision by classifying only when models consistent with the training data predict the same output (Khani et al., 2016).

**Uncertainty Quantification (classification + regression).** It was further shown by (Lakshminarayanan et al., 2017; Zaoui et al., 2020) that deep model ensembles (i.e., a collection of multiple models trained with different hyper-parameters until convergence) can provide state-of-the-art uncertainty quantification, a task closely related to selective prediction. This however raises the need to train multiple models from scratch. To reduce the cost of training multiple models, (Gal & Ghahramani, 2016) proposed abstention based on the variance statistics from several dropout Srivastava et al. (2014) enabled forward passes at test time. Another popular technique for uncertainty quantification, especially for regression and time series forecasting, is given by directly modeling the output distribution (Alexandrov et al., 2019) in a parametric fashion. Training with a parametric output distribution however can lead to additional training instability, often requiring extensive hyper-parameter tuning and distributional assumptions. On the other hand, our approach does not require any architecture or other training-time modifications. Finally, we note that selective prediction and uncertainty are also strongly related to the field of automated model evaluation which relies on the construction a proximal prediction pipeline of the testing performance without the presence of ground-truth labels (Peng et al., 2023; 2024).

**Training Dynamics Approaches (classification).** Checkpoint and snapshot ensembles (Huang et al., 2017; Chen et al., 2017) constitute the first usage of training dynamics to boost model utility. Our work is closest in spirit to recent work on dataset cartography (Swayamdipta et al., 2020) which relies on using training dynamics from an example difficulty viewpoint by considering the variance of logits. However, their approach does not consider selective prediction and further requires access to true label information (which is not available in the selective prediction setting). Recent work on out-of-distribution detection (Adila & Kang, 2022), a closely related yet distinct application scenario from selective prediction, harness similar training dynamics based signals.

---

[1]Under miscalibration, a model's prediction frequency of events does not match the true observed frequency of events.

**Example Difficulty.** A related line of work to selective prediction is identifying *difficult* examples, or how well a model can generalize to a given unseen example. Recent work Jiang et al. (2020) has demonstrated that the probability of predicting the ground truth label with models trained on data sub-samples of different sizes can be estimated via a per-instance empirical consistency score. Unlike our approach, however, this requires training a large number of models. Example difficulty can also be quantified through the lens of a forgetting event Toneva et al. (2018) in which the example is misclassified after being correctly classified. Instead, the metrics that we introduce in Section 3, are based on the disagreement of the label at each checkpoint with the final predicted label. Other approaches estimate the example difficulty by: prediction depth of the first layer at which a $k$-NN classifier correctly classifies an example (Baldock et al., 2021), the impact of quantization and compression on model predictions of a given sample (Hooker et al., 2019), and estimating the leave-one-out influence of each training example on the accuracy of an algorithm by using influence functions (Feldman & Zhang, 2020). Closest to our method, the work of Agarwal et al. (2020) utilizes gradients of intermediate models during training to rank examples by difficulty. In particular, they average pixel-wise variance of gradients for each given input image. Notably, this approach is more costly and less practical than our approach and also does not study the utility/coverage trade-off which is of paramount importance to selective prediction.

**Disagreement** Our **SPTD** method heavily relies on the presence of disagreements between intermediate models. Past work on (dis-)agreement has studied the connection between generalization and disagreement of full SGD runs (Jiang et al., 2021) as well as correlations between in-distribution and out-of-distribution agreement across models (Baek et al., 2022).

## 3 Selective Prediction via Neural Network Training Dynamics

We now introduce our selective prediction algorithms based on neural network training dynamics. We start by presenting a motivating example showcasing the effectiveness of analyzing training trajectories for a linear classification problem. Following this, we formalize our selective prediction scoring rule based on training-time prediction disagreements. We refer to the class of methods we propose as **SPTD**.

### 3.1 Method Intuition: Prediction Disagreements Generalize Softmax Response

Stochastic iterative optimization procedures, such as Stochastic Gradient Descent (SGD), yield a sequence of models that is iteratively derived by minimizing a loss function $\ell(\cdot, \cdot)$ on a randomly selected mini-batch $(\boldsymbol{X}_i, \boldsymbol{y}_i)$ from the training set. The iterative update rule can be expressed as follows

$$\boldsymbol{\theta}_{t+1} = \boldsymbol{\theta}_t - \nu \frac{\partial \ell(f(\boldsymbol{X}_i), \boldsymbol{y}_i)}{\partial \boldsymbol{\theta}_t} \tag{10}$$

where the learning rate $\nu$ controls the speed of optimization and $t \in \{1, \ldots, T\}$ represents a particular time-step during the optimization process.

Current methods for selective prediction disregard the properties of this iterative process and only rely on the final set of parameters $\boldsymbol{\theta}_T$. However, the optimization trajectories contain information that we can use to determine prediction reliability. In particular, on hard optimization tasks, the presence of stochasticity from SGD and the potential ambiguity of the data often leads to noisy optimization behavior. As a result, intermediate predictions produced over the course of training might widely disagree in what the right prediction would be for a given data point. Our class of selective prediction approaches explicitly make use of these training dynamics by formalizing rejection scores based on the observed frequency of prediction disagreements with the final model throughout training.

To illustrate and reinforce this intuition that training dynamics contain meaningfully more useful information for selective prediction than the final model, we present a synthetic logistic regression example. First, we generate a mixture of two Gaussians each consisting of 1000 samples: $D = \{(\boldsymbol{x}_i, 0)\}_{i=1}^{1000} \cup \{(\boldsymbol{x}_j, 1)\}_{j=1}^{1000}$ where $\boldsymbol{x}_i \sim \mathcal{N}(\begin{bmatrix} a & 0 \end{bmatrix}^\top, \boldsymbol{I})$ and $\boldsymbol{x}_j \sim \mathcal{N}(\begin{bmatrix} -a & 0 \end{bmatrix}^\top, \boldsymbol{I})$. Note that $a$ controls the distance between the two 2-dimensional Gaussian clusters, allowing us to specify the difficulty of the learning task. Then, we train a

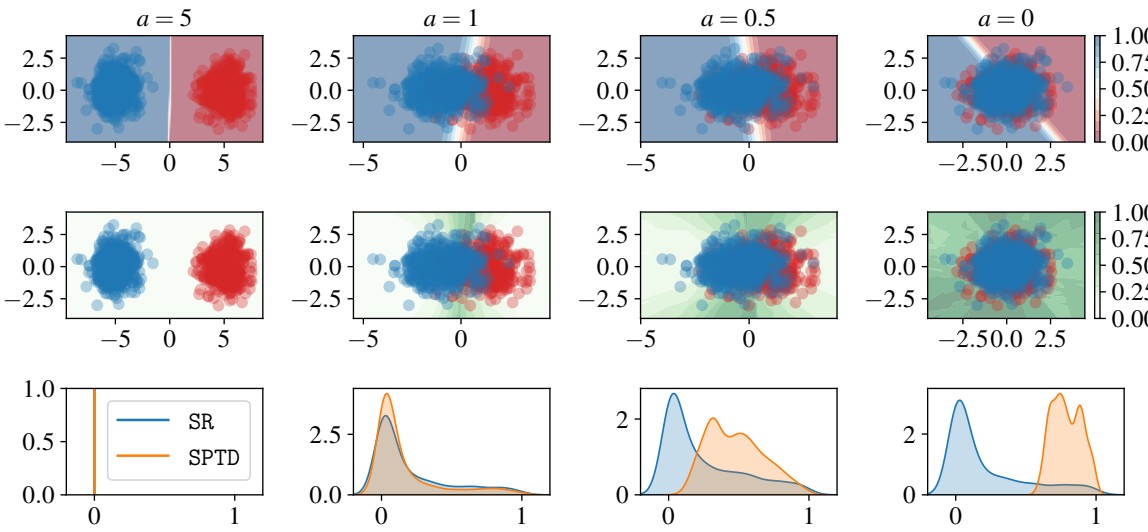

Figure 2: **Synthetic example of anomaly scoring based on SR vs SPTD**. The first row shows a test set from the generative Gaussian model as well as the learned decision boundary separating the two Gaussians. For small $a$, the decision boundary is overconfident. The second row shows the same data set but instead depicts the scores yielded by applying **SPTD** to the full domain. **SPTD** highlights rightful rejection regions more clearly than the overconfident **SR** score: larger regions are flagged as exhibiting noisy training dynamics (with stronger shades of green indicating stronger disagreement) as $a \to 0$. The bottom row shows the distribution of the **SR** and **SPTD** scores, clearly showing that **SPTD** leads to improved uncertainty under stronger label ambiguity.

linear classification model using SGD for 1000 epochs for each $a \in \{0, 0.5, 1, 5\}$. Finally, we compute both the softmax response score (**SR**) score, the typical baseline for selective classification, as well as our **SPTD** score (details in Section 3.2).

We showcase the results from this experiment in Figure 2. We see that if the data is linearly separable ($a = 5$) the learned decision boundary is optimal and the classifier's built-in confidence score **SR** reflects well-calibrated uncertainty. Moreover, the optimization process is stable as **SPTD** yields low scores over the full domain. However, as we move the two Gaussians closer together (i.e., by reducing $a$) we see that the **SR** baseline increasingly suffers from overconfidence: large parts of the domain are very confidently classified as either 0 (red) or 1 (blue) with only a small ambiguous region (white). However, the optimization trajectory is highly unstable with the decision boundary changing abruptly between successive optimization steps. **SPTD** identifies the region of datapoints exhibiting large prediction disagreement due to this training instability and correctly rejects them (as those are regions also subject to label ambiguity in this case). In summary, we observe that **SPTD** provides improved uncertainty quantification in ambiguous classification regions (which induce training instability) and reduces to the **SR** solution as the classification task becomes easier. Hence, we expect **SPTD** to generalize **SR** performance, which is supported by this logistic regression experiment.

## 3.2 Method Overview: Measuring Prediction Instability During Training

We proceed to describe the statistics we collect from intermediate checkpoints which we later devise our scores for deciding which inputs to reject on. The objective of these statistics is to capture how unstable the prediction for a datapoint was over the training checkpoints. Let $[f_1, f_2, \ldots, f_T]$ be a sequence of intermediate checkpoints, and $\mathcal{D} = D_{\text{train}} \cup D_{\text{test}}$ be the set of all data points. We define a prediction disagreement score at time $t \in \{1, \ldots, T\}$ as some function $a_t : \mathcal{X} \to \mathbb{R}^+$ with $a_t(\boldsymbol{x}) = 0$ if $f_t(\boldsymbol{x}) = f_T(\boldsymbol{x})$. Note that the exact $a_t(\cdot)$ we use depends on the problem domain (classification vs regression) and we define our choices below.

| **Algorithm 1: SPTD** for classification | **Algorithm 2: SPTD** for regression |
|---|---|
| **Require:** Intermediate models $[f_1, \ldots, f_T]$, query point $\boldsymbol{x}$, weighting parameter $k \in [0, \infty)$. | **Require:** Intermediate models $[f_1, \ldots, f_T]$, query point $\boldsymbol{x}$, weighting parameter $k \in [0, \infty)$. |
| 1: **for** $t \in [T]$ **do** | 1: **for** $t \in [T]$ **do** |
| 2:   **if** $f_t(\boldsymbol{x}) = f_T(\boldsymbol{x})$ **then** $a_t \leftarrow 0$ **else** $a_t \leftarrow 1$ | 2:   $a_t \leftarrow \|f_t(\boldsymbol{x}) - f_T(\boldsymbol{x})\|$ |
| 3:   $v_t \leftarrow (\frac{t}{T})^k$ | 3:   $v_t \leftarrow (\frac{t}{T})^k$ |
| 4: **end for** | 4: **end for** |
| 5: $g \leftarrow \sum_t v_t a_t$ | 5: $g \leftarrow \sum_t v_t a_t$ |
| 6: **if** $g \leq \tau$ **then** $f(\boldsymbol{x}) = f_T(\boldsymbol{x})$ **else** $f(\boldsymbol{x}) = \bot$ | 6: **if** $g \leq \tau$ **then** $f(\boldsymbol{x}) = f_T(\boldsymbol{x})$ **else** $f(\boldsymbol{x}) = \bot$ |

In the following, when conditioning on $\boldsymbol{x}$ is understood from context, we drop the explicit dependence on $\boldsymbol{x}$ and write $a_t$.

For a fixed data point $\boldsymbol{x}$, our approach takes a given sequence of prediction disagreements $[a_1, \ldots, a_T]$ and associates a weight $v_t$ to each disagreement $a_t$ to capture how severe a disagreement at step $t$ is. To derive this weighting we ask: How indicative of $\boldsymbol{x}$ being incorrectly classified is a disagreement at step $t$? Related work in the example difficulty literature (see Section 2.1 for details) found that easy-to-optimize samples are learned early in training and converge faster. While prior work specifically derived these convergence insights for training points only, the novelty of our method is to show such conclusions for training points also generalize to test points. Hence, we propose to use the weighting $v_t = (\frac{t}{T})^k$ for $k \in [0, \infty)$ to penalize late prediction disagreements as more indicative of a test point we will not predict correctly on. With this weighting, our methods compute a weighted sum of the prediction disagreements, which effectively forms our selection function $g(\cdot)$:

$$g(\boldsymbol{x}) = \sum_t v_t a_t(\boldsymbol{x}) \tag{11}$$

**Instability for Classification.** For discrete prediction problems (i.e., classification) we define the label disagreement score as $a_t = 1 - \delta_{f_t(\boldsymbol{x}), f_T(\boldsymbol{x})}$ where $\delta$ is the Dirac-delta function: $a_t$ is hence 1 if the intermediate prediction $f_t$ at checkpoint $t$ disagrees with the final prediction $f_T$ for $\boldsymbol{x}$, else 0. The resulting algorithm using this definition of $a_t$ for classification is given in Algorithm 1. We remark that continuous metrics such as the maximum softmax score, the predictive entropy (i.e., the entropy of the predictive distribution $f(y|\boldsymbol{x})$), or the gap between the two most confident classes could be used as alternate measures for monitoring stability (see Appendix A for a discussion). However, these measures only provide a noisy proxy and observing a discrete deviation in the predicted class provides the most direct signal for potential mis-classification.

**Instability for Regression.** One key advantage of our method over many previous ones is that it is applicable to *any* predictive model, including regression. Here, we propose the following prediction disagreement score measuring the distance of intermediate predictions to the final model's prediction: $a_t = \|f_t(\boldsymbol{x}) - f_T(\boldsymbol{x})\|$.[2] The resulting algorithm using this definition of $a_t$ for regression is given in Algorithm 2. We again highlight the fact that Algorithm 2 only differs from Algorithm 1 in the computation of the prediction disagreement $a_t$ (line 2 highlighted in both algorithms).

**Instability for Time Series Prediction.** We can further generalize the instability sequence used for regression to time series prediction problems by computing the regression score for all time points on the prediction horizon. In particular, we compute $a_{t,r} = \|f_t(\boldsymbol{x})_r - f_T(\boldsymbol{x})_r\|$ for all $r \in \{1, \ldots, R\}$. Recall that for time series problems $f_t(\boldsymbol{x})$ returns a vector of predictions $y \in \mathbb{R}^R$ and we use the subscript $r$ on $f_t(\boldsymbol{x})_r$ to denote the vector indexing operation. Our selection function is then given by computing Equation 11 for each $r$ and summing up the instabilities over the prediction horizon: $g(\boldsymbol{x}) = \sum_r \sum_t v_t a_{t,r}(\boldsymbol{x})$. The full algorithm therefore shares many conceptual similarities with Algorithm 2 and we provide the detailed algorithm as part of Algorithm 3. Note that the presented generalization for time series is applicable to any setting in

---

[2]We explored a more robust normalization by averaging predictions computed over the last $l$ checkpoints: $a_t = \|f_t(\boldsymbol{x}) - \frac{1}{n}\sum_{c \in \{T-l, T-l+1, \ldots, T\}} f_c(\boldsymbol{x})\|$. Across many $l$, we found the obtained results to be statistically indistinguishable from the results obtained by normalizing w.r.t. the last checkpoint $f_T$.

---

**Algorithm 3: SPTD** for time series forecasting

---

**Require:** Intermediate models $[f_1, \ldots, f_T]$, query point $\boldsymbol{x}$, weighting $k \in [0, \infty)$, prediction horizon $R$.

 1: **for** $t \in [T]$ **do**
 2:   **for** $r \in [R]$ **do**
 3:     $a_{t,r} \leftarrow ||f_t(\boldsymbol{x})_r - f_T(\boldsymbol{x})_r||$
 4:   **end for**
 5:   $v_t \leftarrow (\frac{t}{T})^k$
 6: **end for**
 7: $g \leftarrow \sum_r \sum_t v_t a_{t,r}$
 8: **if** $g \leq \tau$ **then** $f(\boldsymbol{x}) = L$ **else** $f(\boldsymbol{x}) = \perp$

---

which the variability of predictions can be computed. As such, this formalism can extend to application scenarios beyond time series prediction such as object detection or segmentation.

## 3.3 Selective Prediction and Forging

While our **SPTD** method is primarily motivated from the example difficulty view point, we remark that the scores **SPTD** computes to decide which points to reject can be derived from multiple different perspectives. To showcase this, we provide a formal treatment on the connection between selective classification and forging (Thudi et al., 2022), which ultimately leads to the same selection function $g(\cdot)$ as above.

Previous work has shown that running SGD on different datasets could lead to the same final model (Hardt et al., 2016; Bassily et al., 2020; Thudi et al., 2022). For example, this is intuitive when two datasets were sampled from the same distribution. We would then expect that training on either dataset should not significantly affect the model returned by SGD. For our selective prediction problem, this suggests an approach to decide which points the model is likely to predict correctly on: identify the datasets that it could have been trained on (in lieu of the training set it was actually trained on). Any point from the datasets the model could have trained on would then be likely to be predicted on correctly by the model. Recent work on forging Thudi et al. (2022) solves this problem of identifying datasets the model could have trained on by brute-force searching through different mini-batches to determine if a mini-batch in the alternative dataset can be used to reproduce one of the original training steps. Even then, this is only a sufficient condition to show a datapoint could have plausibly been used to train: if the brute-force fails, it does not mean the datapoint could not have been used to obtain the final model. As an alternative, we propose to instead characterize the optimization behaviour of training on a dataset as a probabilistic necessary condition, i.e, a condition most datapoints that were (plausibly) trained on would satisfy based on training dynamics. Our modified hypothesis is then that the set of datapoints we optimized for (which contains the forgeable points) coincides significantly with the set of points the model predicts correctly on.

### 3.3.1 A Framework for Being Optimized

In this section we derive an upper-bound on the probability that a datapoint could have been used to obtain the model's checkpointing sequence. This yields a probabilistically necessary (though not sufficient) characterization of the points we explicitly optimized for. This bound, and the variables it depends on, informs what we characterize as "optimizing" for a datapoint, and, hence, our selective classification methods.

Let us denote the set of all datapoints as $\mathcal{D}$, and let $D \subset \mathcal{D}$ be the training set. We are interested in the setting where a model $f$ is plausibly sequentially trained on $D$ (e.g., with stochastic gradient descent). We thus also have access to a sequence of $T$ intermediate states for $f$, which we denote $[f_1, \ldots, f_T]$. In this sequence, note that $f_T$ is exactly the final model $f$.

Now, let $p_t$ represent the random variable for outputs on $D$ given by an intermediate model $f_t$ where the outputs have been binarized: we have 0 if the output agrees with the final prediction and 1 if not. In other words, $p_t$ is the distribution of labels given by first drawing $\boldsymbol{x} \sim D$ and then outputting $1 - \delta_{f_t(\boldsymbol{x}), f_T(\boldsymbol{x})}$ where $\delta$ denotes the Dirac delta function. Note that we always have both a well-defined mean and variance

for $p_t$ as it is bounded. Furthermore, we always have the variances and expectations of $\{p_t\}$ converge to 0 with increasing $t$: as $p_T = 0$ always and the sequence is finite convergence trivially occurs. To state this formally, let $v_t = \mathbb{V}_{\boldsymbol{x} \sim D}[p_t]$ and let $e_t = \mathbb{E}_{\boldsymbol{x} \sim D}[p_t]$ denote the variances and expectations over points in $D$. In particular, we remark that $e_T = 0$, $v_T = 0$, so both $e_t$ and $v_t$ converge. More formally, for all $\epsilon > 0$ there exists an $N \in \{1, \ldots, T\}$ such that $v_t < \epsilon$ for all $t > N$. Similarly, for all $\epsilon > 0$ there exists a (possibly different) $N \in \{1, \ldots, T\}$ such that $e_t < \epsilon$ for all $t > N$.

However, the core problem is that we do not know how this convergence in the variance and expectation occurs. More specifically, if we knew the exact values of $e_t$ and $v_t$, we could use the following bound on the fraction of training data points producing a given $[a_1, \cdots, a_t]$ as a reject option for points that are not optimized for. We consequently introduce the notation $[a_1, \ldots, a_T]$ where $a_t = 1 - \delta_{f_t(\boldsymbol{x}), f_T(\boldsymbol{x})}$ which we call the "label disagreement (at $t$)". Note that the $a_t$ are defined with respect to a given input, while $p_t$ represent the distribution of $a_t$ over all inputs in $D$.

**Lemma 1.** *Given a datapoint $\boldsymbol{x}$, let $\{a_1, \ldots, a_T\}$ where $a_t = 1 - \delta_{f_t(\boldsymbol{x}), f_T(\boldsymbol{x})}$. Assuming not all $a_t = e_t$ then the probability $\boldsymbol{x} \in D$ is $\leq \min_{v_t \ s.t \ a_t \neq e_t} \frac{v_t}{|a_t - e_t|^2}$.*

*Proof.* By Chebyshev's inequality we have the probability of a particular sequence $\{a_1, \ldots, a_T\}$ occurring for a training point is $\leq \frac{v_t}{|a_t - e_t|^2}$ for every $t$ (a bound on any of the individual $a_t$ occurring as that event is in the event $|p_t - e_t| \geq |a_t - e_t|$ occurs). By taking the minimum over all these upper-bounds we obtain our upper-bound. □

We do not guarantee Lemma 1 is tight. Though we do take a minimum to make it tighter, this is a minimum over inequalities all derived from Chebyshev's inequality[3]. Despite this potential looseness, using the bound from Lemma 1, we can design a naïve selective classification protocol based on the "optimized = correct (often)" hypothesis and use the above bound on being a plausible training datapoint as our characterization of optimization; for a test input $\boldsymbol{x}$, if the upper-bound on the probability of being a datapoint in $D$ is lower than some threshold $\tau$ reject, else accept. However, the following question prevents us from readily using this method: *How do $\mathbb{E}[p_t]$ and $\mathbb{V}[p_t]$ evolve during training?*

To answer this question, we propose to examine how the predictions on plausible training points evolve during training. Informally, the evolution of $\mathbb{E}[p_t]$ represents knowing how often we predict the final label at step $t$, while the evolution of $\mathbb{V}[p_t]$ represents knowing how we become more consistent as we continue training. Do note that the performance of this optimization-based approach to selective classification will depend on how unoptimized incorrect test points are. In particular, our hypothesis is that incorrect points often appear sufficiently un-optimized, yielding distinguishable patterns for $\mathbb{E}[p_t]$ and $\mathbb{V}[p_t]$ when compared to optimized points. We verify this behavior in Section 4 where we discuss the distinctive label evolution patterns of explicitly optimized, correct, and incorrect datapoints.

### 3.3.2 Last Disagreement Model Score For Discrete Prediction ($s_{\text{MAX}}$)

Here, we propose a selective classification approach based on characterizing optimizing for a datapoint based off of Lemma 1. Recall the bound given in Lemma 1 depends on expected values and variances for the $p_t$ (denoted $e_t$ and $v_t$ respectively). In Section 4 we observe that $e_t$ quickly converge to 0, and so by assuming $e_t = 0$ always[4] the frequentist bound on how likely a datapoint is a training point becomes $\min_{t \ s.t \ a_t = 1} \frac{v_t}{|a_t - e_t|^2} = \min_{t \ s.t \ a_t = 1} v_t$. Using this result for selective classification, we would impose acceptance if $\min_{t \ s.t \ a_t = 1} v_t \geq \tau$. Moreover, in Section 4, we further observe that $v_t$ monotonically decreases in a convex manner (after an initial burn-in phase). Hence, imposing $\min_{t \ s.t \ a_t = 1} v_t \geq \tau$ simply imposes a last checkpoint that can have a disagreement with the final prediction.

Based on these insights, we propose the following selective classification score: $s_{\max} = \max_{t \ s.t \ a_t = 1} \frac{1}{v_t}$. Note that this score directly follows from the previous discussion but flips the thresholding direction from $\min_{t \ s.t \ a_t = 1} v_t \geq \tau$ to $\max_{t \ s.t \ a_t = 1} \frac{1}{v_t} \leq \tau$ for consistency with the anomaly scoring literature (Ruff et al., 2018). Finally, we choose to approximate the empirical trend of $v_t$ as observed in Section 4 with $v_t = 1 - t^k$

---

[3]One could potentially use information about the distribution of points not in $D$ to refine this bound.
[4]We tried removing this assumption and observed similar performance.

for $k \in [1, \infty)$. Based on the choice of $k$, this approximation allows us to (i) avoid explicit estimation of $v_t$ from validation data; and (ii) enables us to flexibly specify how strongly we penalize model disagreements late in training.

Hence, our first algorithm for selective classification is:

1. Denote $L = f_T(\boldsymbol{x})$, i.e. the label our final model predicts.

2. If $\exists t \ s.t \ a_t = 1$ then compute $s_{\max} = \max_{t \ s.t \ a_t=1} \frac{1}{v_t}$ as per the notation in Section 3.3.1 (i.e $a_t = 1$ iff $f_t(x) \neq L$), else accept $\boldsymbol{x}$ with prediction $L$.

3. If $s_{\max} \leq \tau$ accept $\boldsymbol{x}$ with prediction $L$, else reject ($\perp$).

Note once again, as all our candidate $\frac{1}{v_t}$ increase, the algorithm imposes a last intermediate model which can output a prediction that disagrees with the final prediction: hereafter, the algorithm must output models that consistently agree with the final prediction.

### 3.3.3 Overall Disagreement Model Score ($s_{\mathsf{SUM}}$)

Note that the previous characterization of optimization, defined by the score $s_{\mathrm{MAX}}$, could be sensitive to stochasticity in training and hence perform sub-optimally. That is, the exact time of the last disagreement, which $s_{\mathrm{MAX}}$ relies on, is subject to high noise across randomized training runs. In light of this potential limitation we propose the following "summation" algorithm which computes a weighted sum over training-time disagreements to get a more consistent statistic. Do note that typically to get a lower-variance statistic one would take an average, but multiplying by scalars can be replaced by correspondingly scaling the threshold we use. Hence, our proposed algorithm is:

1. Denote $L = f_T(\boldsymbol{x})$, i.e. the label our final model predicts.

2. If $\exists t \ s.t \ a_t = 1$, compute $s_{\mathrm{sum}} = \sum_{t=1}^{T} \frac{a_t}{v_t}$, else accept $\boldsymbol{x}$ with prediction $L$.

3. If $s_{\mathrm{sum}} \leq \tau$ accept $\boldsymbol{x}$ with prediction $L$, else reject ($\perp$).

Recalling our previous candidates for $v_t$, we have the $s_{\mathrm{SUM}}$ places higher weight on late disagreements. This gives us a biased average of the disagreements which intuitively approximates the expected last disagreement but now is less susceptible to noise. More generally, this statistic allows us to perform selective classification by utilizing information from all the disagreements during training. In Appendix B.2.7, we experimentally show that $s_{\mathrm{SUM}}$ leads to more robust selective classification results compared to $s_{\mathrm{MAX}}$. **We remark that the sum score $s_{\mathsf{SUM}}$ corresponds exactly to our score $g(\cdot)$ proposed as part of SPTD (recall Equation 11 from Section 3.2), showcasing the strong connection of our method to forging.**

## 4 Empirical Evaluation

We present a comprehensive empirical study demonstrating the effectiveness of **SPTD** across domains. Our results show that computing and thresholding the proposed weighted instability score from **SPTD** provides a strong score for selective classification, regression, and time series prediction.

### 4.1 Classification

**Key Research Goals.** As part of our experiments we:

- Study the accuracy/coverage trade-off with comparison to past work, showing that **SPTD** outperforms existing work.

- Present exemplary training-dynamics-derived label evolution curves for individual examples from all datasets.

- Examine our method's sensitivity to the checkpoint selection strategy and the weighting parameter $k$.

- Evaluate the detection performance of out-of-distribution and adversarial examples, showing that **SPTD** can be applied beyond the i.i.d. assumption of selective prediction.

- Provide a detailed cost vs performance tradeoff of **SPTD** and competing selective prediction methods.

- Analyze distributional training dynamics patterns of both correct and incorrect data points, the separation of which enables performative selective classification.

**Datasets & Training.**    We evaluate **SPTD** on image dataset benchmarks that are common in the selective classification literature: CIFAR-10/CIFAR-100 (Krizhevsky et al., 2009), StanfordCars (Krause et al., 2013), and Food101 (Bossard et al., 2014). For each dataset, we train a deep neural network following the ResNet-18 architecture (He et al., 2016) and checkpoint each model after processing 50 mini-batches of size 128. All models are trained over 200 epochs (400 epochs for StanfordCars) using the SGD optimizer with an initial learning rate of $10^{-2}$, momentum 0.9, and weight decay $10^{-4}$. Across all datasets, we decay the learning rate by a factor of 0.5 in 25-epoch intervals.

**Baselines.**    We compare our method (**SPTD**) to common SC techniques previously introduced in Section 2: Softmax Response (**SR**) and Self-Adaptive Training (**SAT**). Based on recent insights from Feng et al. (2023), we (i) train **SAT** with additional entropy regularization[5]; and (ii) derive **SAT**'s score by applying Softmax Response (**SR**) to the underlying classifier (instead of thresholding the abstention class). We refer to this method as **SAT+ER+SR**. We do not include results for SelectiveNet, Deep Gamblers, or Monte-Carlo Dropout as previous works (Huang et al., 2020; Feng et al., 2023) have shown that **SAT+ER+SR** strictly dominates these methods. In contrast to recent SC works, we do however include results with Deep Ensembles (**DE**) (Lakshminarayanan et al., 2017), a relevant baseline from the uncertainty quantification literature. Our hyper-parameter tuning procedure is documented in Appendix B.1.

**Accuracy/Coverage Trade-off.**    Consistent with standard evaluation schemes for selective classification, our main experimental results examine the accuracy/coverage trade-off of **SPTD**. We present our performance results with comparison to past work in Table 1 where we demonstrate **SPTD**'s effectiveness on CIFAR-10, CIFAR-100, StanfordCars, and Food101. We document the results obtained by **SPTD**, **SAT**, **SR**, and **DE** across the full coverage spectrum. We see that **SPTD** outperforms both **SAT** and **SR** and performs similarly as **DE**. To further boost performance across the accuracy/coverage spectrum, we combine **SPTD** and **DE** by applying **SPTD** on each ensemble member from **DE** and then average their scores. More concretely, we estimate $\textbf{DE+SPTD} = \frac{1}{m} \sum_{m=1}^{M} \textbf{SPTD}_m$ where $\textbf{SPTD}_m$ computes $g$ on each ensemble member $m \in [M]$. This combination leads to new state-of-the-art selective classification performance and showcases that **SPTD** can be flexibly applied on top of established training pipelines. Further evidence towards this flexibility is provided in Appendix B.2.3 where we show that applying **SPTD** on top of **SAT** also improves selective prediction performance.

**Individual Evolution Plots.**    To analyze the effectiveness of our disagreement metric proposed in Section 3, we examine the evolution curves of our indicator variable $a_t$ for individual datapoints in Figure 3. In particular, for each dataset, we present the most stable and the most unstable data points from the test sets and plot the associated label disagreement metric $a_t$ over all checkpoints. We observe that easy-to-classify examples only show a small degree of oscillation while harder examples show a higher frequency of oscillations, especially towards the end of training. This result matches our intuition: our model should produce correct decisions on data points whose prediction is mostly constant throughout training and should reject data points for which intermediate models predict inconsistently. Moreover, as depicted in Figure 4, we also show that our score $g(\cdot)$ yields distinct distributional patterns for both correctly and incorrectly classified points. This separation enables strong coverage/accuracy trade-offs via our thresholding procedure.

---

[5]This entropy regularization step is designed to encourage the model to be more confident in its predictions.

| | Coverage | SR | SAT+ER+SR | DE | SPTD | DE+SPTD |
|---|---|---|---|---|---|---|
| *CIFAR-10* | 100 | **92.9 (±0.0)** | **92.9 (±0.0)** | **92.9 (±0.0)** | **92.9 (±0.0)** | 92.9 (±0.1) |
| | 90 | 96.4 (±0.1) | 96.3 (±0.1) | **96.8 (±0.1)** | 96.5 (±0.0) | 96.7 (±0.1) |
| | 80 | 98.1 (±0.1) | 98.1 (±0.1) | **98.7 (±0.0)** | 98.4 (±0.1) | 98.8 (±0.1) |
| | 70 | 98.6 (±0.2) | 99.0 (±0.1) | **99.4 (±0.1)** | 99.2 (±0.0) | 99.5 (±0.0) |
| | 60 | 98.7 (±0.1) | 99.4 (±0.0) | 99.6 (±0.1) | **99.6 (±0.2)** | 99.8 (±0.0) |
| | 50 | 98.6 (±0.2) | 99.7 (±0.1) | 99.7 (±0.1) | 99.8 (±0.0) | 99.9 (±0.0) |
| | 40 | 98.7 (±0.0) | 99.7 (±0.0) | 99.8 (±0.0) | 99.8 (±0.1) | 100.0 (±0.0) |
| | 30 | 98.5 (±0.0) | 99.8 (±0.0) | 99.8 (±0.0) | 99.8 (±0.1) | 100.0 (±0.0) |
| | 20 | 98.5 (±0.1) | 99.8 (±0.1) | 99.8 (±0.0) | **100.0 (±0.0)** | 100.0 (±0.0) |
| | 10 | 98.7 (±0.0) | 99.8 (±0.1) | 99.8 (±0.1) | **100.0 (±0.0)** | 100.0 (±0.0) |
| *CIFAR-100* | 100 | **75.1 (±0.0)** | **75.1 (±0.0)** | **75.1 (±0.0)** | **75.1 (±0.0)** | 75.1 (±0.0) |
| | 90 | 78.2 (± 0.1) | 78.9 (± 0.1) | 80.2 (± 0.0) | 80.4 (± 0.1) | **81.1 (± 0.1)** |
| | 80 | 82.1 (± 0.0) | 82.9 (± 0.0) | 84.7 (± 0.1) | 84.6 (± 0.1) | **85.0 (± 0.2)** |
| | 70 | 86.4 (± 0.1) | 87.2 (± 0.1) | 88.6 (± 0.1) | **88.7 (± 0.0)** | **88.8 (± 0.1)** |
| | 60 | 90.0 (± 0.0) | 90.3 (± 0.2) | 90.2 (± 0.2) | 90.1 (± 0.0) | **90.4 (± 0.1)** |
| | 50 | 92.9 (± 0.1) | 93.3 (± 0.0) | 94.8 (± 0.0) | 94.6 (± 0.0) | **94.9 (± 0.0)** |
| | 40 | 95.1 (± 0.0) | 95.2 (± 0.1) | **96.8 (± 0.1)** | **96.9 (± 0.1)** | **96.9 (± 0.0)** |
| | 30 | 97.2 (± 0.2) | 97.5 (± 0.0) | **98.4 (± 0.1)** | **98.4 (± 0.1)** | **98.5 (± 0.0)** |
| | 20 | 97.8 (± 0.1) | 98.3 (± 0.1) | **99.0 (± 0.0)** | 98.8 (± 0.2) | **99.2 (± 0.1)** |
| | 10 | 98.1 (± 0.0) | 98.8 (± 0.1) | 99.2 (± 0.1) | **99.4 (± 0.1)** | **99.6 (± 0.1)** |
| *Food101* | 100 | **81.1 (±0.0)** | **81.1 (±0.0)** | **81.1 (±0.0)** | **81.1 (±0.0)** | **81.1 (±0.0)** |
| | 90 | 85.3 (±0.1) | 85.5 (±0.2) | 86.2 (±0.1) | 85.7 (±0.0) | **86.7 (±0.0)** |
| | 80 | 87.1 (±0.0) | 89.5 (±0.0) | 90.3 (±0.0) | 89.9 (±0.0) | **91.3 (±0.1)** |
| | 70 | 92.1 (±0.1) | 92.8 (±0.1) | **94.5 (±0.1)** | 93.7 (±0.0) | **94.6 (±0.0)** |
| | 60 | 95.2 (±0.1) | 95.5 (±0.1) | **97.0 (±0.0)** | **97.0 (±0.0)** | **97.0 (±0.0)** |
| | 50 | 97.3 (±0.1) | 97.5 (±0.0) | 98.2 (±0.0) | 98.3 (±0.2) | **98.5 (±0.0)** |
| | 40 | 98.7 (±0.0) | 98.7 (±0.2) | **99.1 (±0.0)** | 99.1 (±0.1) | **99.2 (±0.1)** |
| | 30 | 99.5 (±0.0) | 99.7 (±0.2) | 99.2 (±0.0) | 99.6 (±0.0) | **99.7 (±0.0)** |
| | 20 | 99.7 (±0.1) | 99.7 (±0.2) | **99.9 (±0.1)** | 99.8 (±0.0) | **99.9 (±0.1)** |
| | 10 | 99.8 (±0.0) | 99.8 (±0.1) | **99.9 (±0.1)** | **99.9 (±0.1)** | **99.9 (±0.1)** |
| *StanfordCars* | 100 | **77.6 (±0.0)** | **77.6 (±0.0)** | **77.6 (±0.0)** | **77.6 (±0.0)** | **77.6 (±0.0)** |
| | 90 | 83.0 (±0.1) | 83.0 (±0.2) | **83.7 (±0.1)** | 83.3 (±0.1) | **83.7 (±0.2)** |
| | 80 | 87.6 (±0.0) | 88.0 (±0.1) | 88.7 (±0.1) | **89.3 (±0.0)** | **89.7 (±0.0)** |
| | 70 | 90.8 (±0.0) | 92.2 (±0.1) | 92.4 (±0.1) | **93.6 (±0.0)** | 93.4 (±0.1) |
| | 60 | 93.5 (±0.1) | 95.2 (±0.1) | 95.3 (±0.0) | **96.2 (±0.0)** | **96.3 (±0.0)** |
| | 50 | 95.3 (±0.0) | 96.9 (±0.2) | 96.4 (±0.1) | **97.0 (±0.1)** | **97.1 (±0.3)** |
| | 40 | 96.8 (±0.0) | 97.8 (±0.0) | **97.8 (±0.2)** | **97.8 (±0.1)** | **97.8 (±0.0)** |
| | 30 | 97.5 (±0.1) | 98.2 (±0.2) | 98.6 (±0.0) | 98.2 (±0.2) | **98.9 (±0.0)** |
| | 20 | 98.1 (±0.0) | 98.4 (±0.1) | 98.9 (±0.2) | 98.6 (±0.0) | **99.0 (±0.0)** |
| | 10 | 98.2 (±0.1) | 98.7 (±0.1) | **99.5 (±0.1)** | 98.5 (±0.1) | **99.5 (±0.0)** |

Table 1: **Selective accuracy achieved across coverage levels**. We find that SPTD-based methods out-performs current SOTA error rates across multiple datasets with full-coverage accuracy alignment. Numbers are reported with mean values and standard deviation computed over 5 random runs. **Bold** numbers are best results at a given coverage level across all methods and underlined numbers are best results for methods relying on a single training run only. Datasets are consistent with Feng et al. (2023).

**Checkpoint Weighting Sensitivity.** One important hyper-parameter of our method is the weighting of intermediate predictions. Recall from Section 3 that **SPTD** approximates the expected stability for correctly classified points via a weighting function $v_t = (\frac{t}{T})^k$. In Figure 5 in the Appendix, we observe that **SPTD** is robust to the choice of $k$ and that $k \in [1, 3]$ performs best. At the same time, we find that increasing $k$ too much leads to a decrease in accuracy at medium coverage levels. This result emphasizes that (i) large parts of the training process contain valuable signals for selective classification; and that (ii) early label disagreements arising at the start of optimization should be de-emphasized by our method.

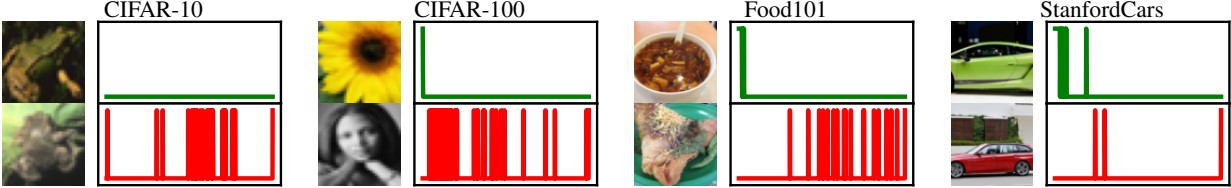

Figure 3: **Most characteristic examples across datasets**. For each dataset, we show the samples with the most stable and most unstable (dis-) agreement with the final label along with their corresponding $a_t$ indicator function. Correct points are predominantly characterized by disagreements early in training while incorrect points change their class label throughout (but importantly close to the end of) training. We provide additional examples from all datasets in Figure 20 in the Appendix.

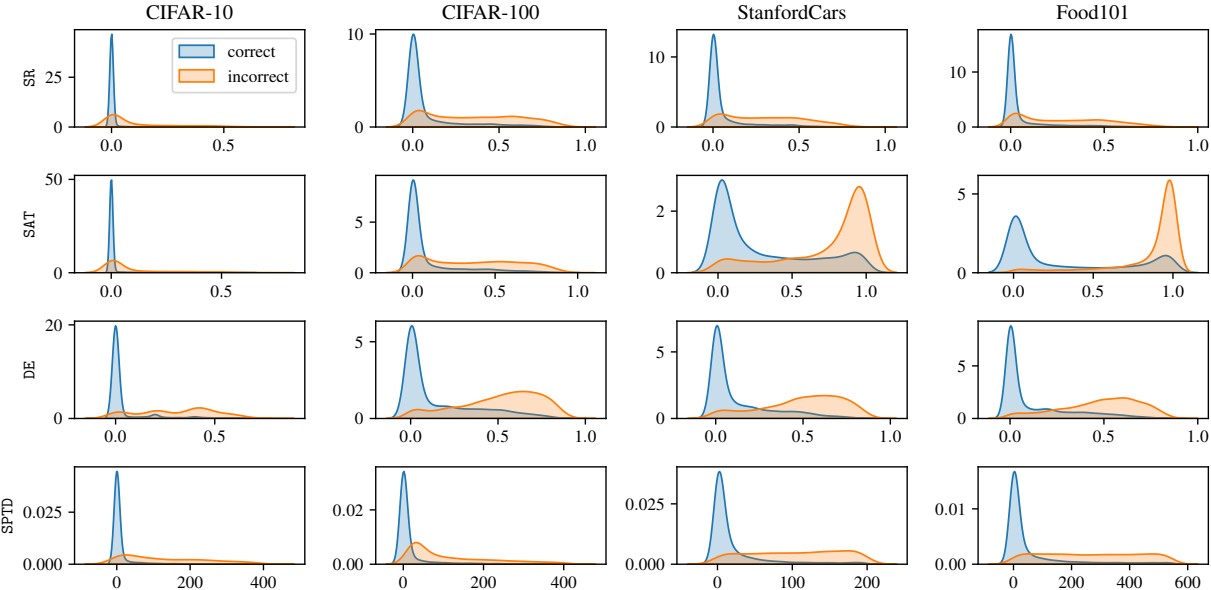

Figure 4: **Distribution of $g$ for different datasets and selective classification methods.** Since all methods are designed to address the selective prediction problem, they all manage to separate correct from incorrect points (albeit at varying success rates). We see that **SPTD** spreads the scores for incorrect points over a wide range with little overlap. We observe that for **SR**, incorrect and correct points both have their mode at approximately the same location which hinders performative selective classification. Although **SAT** and **DE** show larger bumps at larger score ranges, the separation with correct points is weaker as correct points also result in higher scores more often than for **SPTD**.

**Checkpoint Selection Strategy.** The second important hyper-parameter of our method is the checkpoint selection strategy. In particular, to reduce computational cost, we study the sensitivity of **SPTD** with respect to the checkpointing resolution in Figure 6. Our experiments demonstrate favorable coverage/error trade-offs between 25 and 50 checkpoints when considering the full coverage spectrum. However, when considering the high coverage regime in particular (which is what most selective prediction works focus on), even sub-sampling 10 intermediate models is sufficient for SOTA selective classification. Hence, with only observing the training stage, our method's computational overhead reduces to only 10 forward passes at test time when the goal is to reject at most $30\% - 50\%$ of incoming data points. In contrast, **DE** requires to first train $E$ models (with $E = 10$ being a typical and also our particular choice for **DE**) and perform inference on these $E$ models at test time. Further increasing the checkpointing resolution does offer increasingly diminishing returns but also leads to improved accuracy-coverage trade-offs, especially at low coverage.

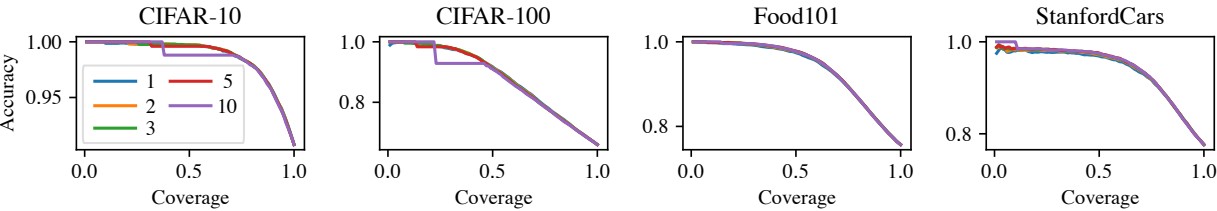

Figure 5: **Coverage/error trade-off of `SPTD` for varying checkpoint weighting $k$ as used in $v_t$.** We observe strong performance for $k \in [1,3]$ across datasets.

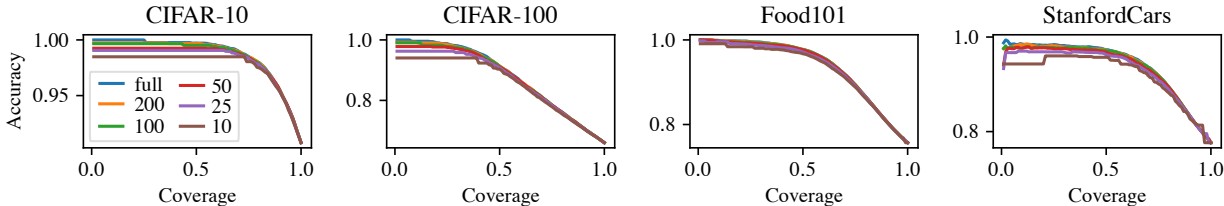

Figure 6: **Coverage/error trade-off of `SPTD` for varying checkpoint counts**. **SPTD** delivers consistent performance independent of the checkpointing resolution at high coverage. At low coverage, a more detailed characterization of training dynamics helps.

**Examining the Convergence Behavior of Training and Test Points.** The effectiveness of **SPTD** relies on our hypothesis that correctly classified points and incorrectly classified points exhibit distinct training dynamics. We verify this hypothesis in Figure 7 where we examine the convergence behavior of the disagreement distributions of correct $(c_t^{\text{tr}})$ / incorrect $(i_t^{\text{tr}})$ training and correct $(c_t^{\text{te}})$ / incorrect $(i_t^{\text{te}})$ test points. We observe that the expected disagreement for both correctly classified training $c_t^{\text{tr}}$ and test points $c_t^{\text{te}}$ points converge to 0 over the course of training. The speed of convergence is subject to the difficulty of the optimization problem with more challenging datasets exhibiting slower convergence in predicted label disagreement. We also see that the variances follow an analogous decreasing trend. This indicates that correctly classified points converge to the final label quickly and fast convergence is strongly indicative of correctness. Furthermore, the overlap suggests that correct test points are more likely to be forgeable as their dynamics look indistinguishable to correct training points (recall Section 3.3 on the connection between our method and forging). In contrast, incorrectly classified points $i_t^{\text{tr}}$ and $i_t^{\text{te}}$ show significantly larger mean and variance levels. This clear separation in distributional evolution patterns across correct and incorrect points leads to strong selective prediction performance in our **SPTD** framework.

**Detection of Out-of-Distribution and Adversarial Examples.** Out-of-distribution (OOD) and adversarial example detection are important disciplines in trustworthy ML related to selective prediction. We therefore provide preliminary evidence in Figure 8 that our method can be used for detecting OOD and adversarial examples. While these results are encouraging, we remark that adversarial and OOD samples are less well defined as incorrect data points and can come in a variety of different flavors (i.e., various kinds of attacks or various degrees of OOD-ness). As such, we strongly believe that future work is needed to determine whether a training-dynamics-based approach to selective prediction can be reliably used for OOD and adversarial sample identification.

**Cost vs Performance Tradeoff.** In Table 2, we report both the time and space complexities for all SC methods at training and test time along with their selective classification performance as per our results in Table 1 and Figure 6. We denote with $E$ the number of **DE** models and with $T$ the number of **SPTD** checkpoints. Although **SR** and **SAT** are the cheapest methods to run, they also perform the poorest at SC. **SPTD** is significantly cheaper to train than **DE** and achieves competitive performance at $T \approx E$. Although **DE+SPTD** is the most expensive model, it also provides the strongest performance.

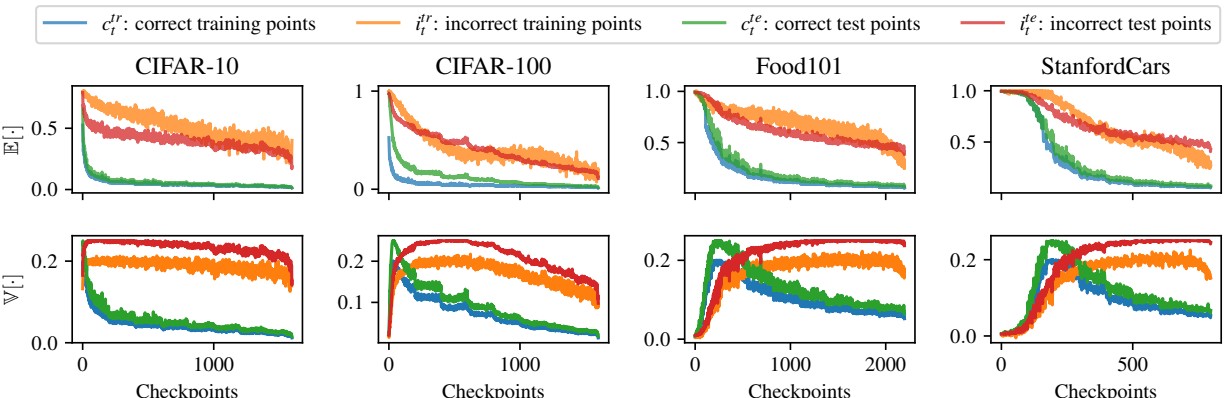

Figure 7: **Monitoring expectations $\mathbb{E}[\cdot]$ and variances $\mathbb{V}[\cdot]$ for correct/incorrect training and test points**. We observe that correctly classified points (cold colors) have both their expectations and variances quickly decreasing to 0 as training progresses. Incorrectly classified points (warm colors) both exhibit large expectations and variances and stay elevated over large periods.

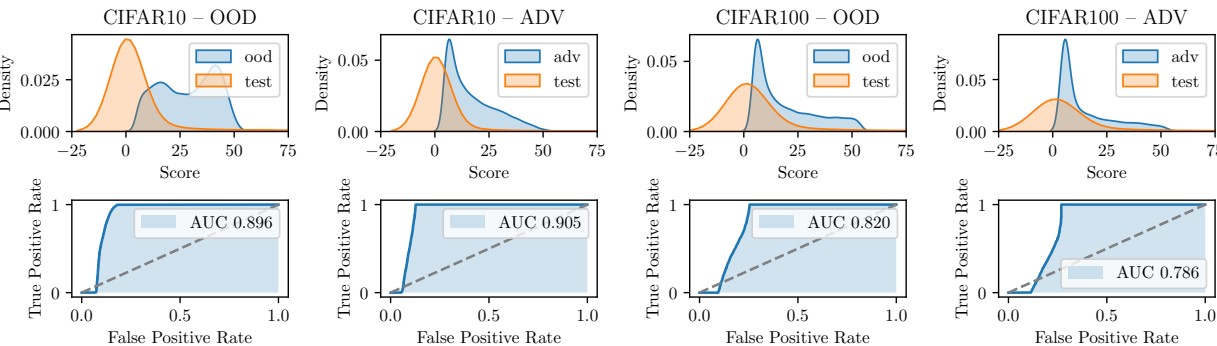

Figure 8: **Performance of SPTD on out-of-distribution (OOD) and adversarial sample detection**. The first row shows the score distribution of the in-distribution CIFAR-10/100 test set vs the SVHN OOD test set or a set consisting of adversarial samples generated via a PGD attack in the final model. The second row shows the effectiveness of a thresholding mechanism by computing the area under the ROC curve. Our score enables separation of anomalous data points from in-distribution test points.

## 4.2 Regression Experiments

**Datasets.** Our experimental suite for regression considers the following datasets: California housing dataset (Pace & Barry, 1997) ($N = 20640$, $D = 8$), the concrete strength dataset (Yeh, 2007) ($N = 1030$, $D = 9$), and the fish toxicity dataset (Ballabio et al., 2019) ($N = 546$, $D = 9$).

**Model Setup & Baselines.** We split all datasets into 80% training and 20% test sets after a random shuffle. Then, we train a fully connected neural network with layer dimensionalities $D \to 10 \to 7 \to 4 \to 1$. Optimization is performed using full-batch gradient descent using the Adam optimizer with learning rate $10^{-2}$ over 200 epochs and weight decay $10^{-2}$. We consider the following baseline methods for rejecting input samples: (i) approximating the predictive variance using deep ensembles (**DE**) (Lakshminarayanan et al., 2017; Zaoui et al., 2020); (ii) SelectiveNet (**SN**) which explicitly optimizes utility given a desired coverage constraint; and (iii) training the model with a Gaussian parametric output distribution (**ODIST**) via maximum likelihood maximization (Alexandrov et al., 2019).

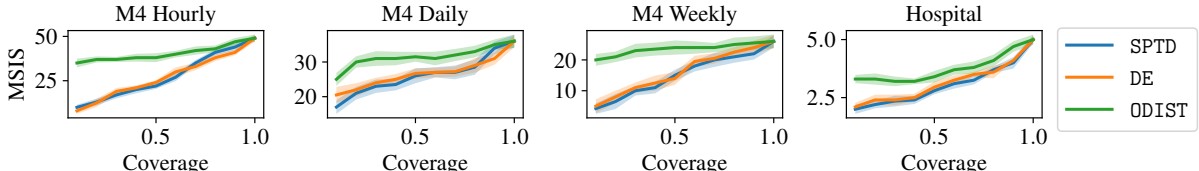

Figure 9: **MSIS/coverage trade-off across various time series prediction datasets**. **SPTD** offers comparable performance to **DE** but provides improved results at low coverage.

| Method | Train Time | Train Space | Inf Time | Inf Space | Rank |
|--------|------------|-------------|----------|-----------|------|
| **SR** | $O(1)$ | $O(1)$ | $O(1)$ | $O(1)$ | 5 |
| **SAT** | $O(1)$ | $O(1)$ | $O(1)$ | $O(1)$ | 4 |
| **DE** | $O(E)$ | $O(E)$ | $O(E)$ | $O(E)$ | =2 |
| **SPTD** | $O(1)$ | $O(T)$ | $O(T)$ | $O(T)$ | =2 |
| **DE+SPTD** | $O(E)$ | $O(ET)$ | $O(ET)$ | $O(ET)$ | 1 |

Table 2: **Cost vs performance tradeoff in terms of training time/space, inference time/space and the performance rank. SPTD** is comparable in performance (at $T \approx E$) and cheaper to train than **DE**. **DE+SPTD** is the most expensive model, but delivers the best performance across datasets.

**Main results.** We document our results in Figure 10. We see that the **ODIST** only delivers subpar results (likely due to mis-calibration) and does not provide a meaningful signal for selective prediction. On the other hand, **DE** and **SPTD** perform comparably with **SPTD** outperforming **DE** at low coverage. We stress again that **SPTD**'s training cost is significantly cheaper than **DE**'s while matching the inference-time cost when sub-sampling a reduced set of checkpoints.

### 4.3 Time Series Experiments

**Datasets.** As part of our time series experiments, we mainly consider the M4 forecasting competition dataset (Makridakis et al., 2020) which contains time series aggregated at various time intervals (e.g., hourly, daily). In addition, we also provide experimentation on the Hospital dataset (Hyndman, 2015).

**Models & Setup.** Our experimentation is carried out using the GluonTS time series framework (Alexandrov et al., 2019). We carry out our experimentation using the DeepAR model (Salinas et al., 2020), a recurrent neural network designed for time series forecasting. We train all models over 200 epochs and evaluate performance using the mean scaled interval score (MSIS) performance metric (Makridakis et al., 2020). Our baselines correspond to the same as presented for regression in Section 4.2: deep ensembles (**DE**), and output parameterization using a Student-t distribution (**ODIST**).

**Main results.** Our time series results are shown in Figure 9 and are consistent with our results for regression: **ODIST** does not provide a meaningful signal for selective prediction while **SPTD** and **DE** perform similarly well. **SPTD** further improves results over **DE** at low converge.

## 5 Conclusion

In this work we have proposed **SPTD**, a selective prediction technique that relies on measuring prediction instability of test points over intermediate model states obtained during training. Our method offers several advantages over previous works. In particular (i) it can be applied to all existing models whose checkpoints were recorded (hence the potential for immediate impact); (ii) it is composable with existing selective prediction techniques; (iii) it can be readily applied to both discrete and real-valued prediction problems; and (iv) it is more computationally efficient than competing ensembling-based approaches. We verified the performance of **SPTD** using an extensive empirical evaluation, leading to new state-of-the-art performance in the field.

Figure 10: $R^2$/**coverage trade-off across various regression datasets**. **SPTD** offers comparable performance to **DE** but provides improved results at low coverage.

Beyond our work, we expect training dynamics information to be useful for identifying and mitigating other open problems in trustworthy machine learning such as (un)fairness, privacy, and model interpretability.

## Acknowledgements

This work was supported by CIFAR (through a Canada CIFAR AI Chair), by NSERC (under the Discovery Program), and by a gift from Ericsson. Anvith Thudi is supported by a Vanier Fellowship. We are also grateful to the Vector Institute's sponsors for their financial support. In particular, we thank Roger Grosse, Chris Maddison, Franziska Boenisch, Natalie Dullerud, Mohammad Yaghini, Sierra Wyllie, Jonas Guan, Michael Zhang, and Tom Ginsberg for fruitful discussions. We would also like to thank the Vector ops and services teams for making the office such a wonderful place to work, and for the (limitless) free hot chocolate.

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

# A   Alternate Metric Choices

We briefly discuss additional potential metric choices that we investigated but which lead to selective classification performance worse than our main method.

## A.1   Jump Score $s_{\mathsf{jmp}}$

We also consider a score which captures the level of disagreement between the predicted label of two successive intermediate models (i.e., how much jumping occurred over the course of training). For $j_t = 0$ iff $f_t(\boldsymbol{x}) = f_{t-1}(\boldsymbol{x})$ and $j_t = 1$ otherwise we can compute the jump score as $s_{\mathsf{jmp}} = 1 - \sum v_t j_t$ and threshold it as in § 3.3.2 and § 3.3.3.

## A.2   Variance Score $s_{\mathsf{var}}$ for Continuous Metrics

We consider monitoring the evolution of continuous metrics that have been shown to be correlated with example difficulty. These metrics include (but are not limited to):

- Confidence (conf): $\max_{c \in \mathcal{Y}} f_t(\boldsymbol{x})$

- Confidence gap between top 2 most confident classes (gap): $\max_{c \in \mathcal{Y}} f_t(\boldsymbol{x}) - \max_{c \neq \hat{y}} f_t(\boldsymbol{x})$

- Entropy (ent): $-\sum_{c=1}^{C} f_t(\boldsymbol{x})_c \log\left(f_t(\boldsymbol{x})_c\right)$

Jiang et al. (2020) show that example difficulty is correlated with confidence and entropy. Moreover, they find that difficult examples are learned later in the training process. This observation motivates designing a score based on these continuous metrics that penalises changes later in the training process more heavily. We consider the maximum softmax class probability known as confidence, the negative entropy and the gap between the most confident classes for each example instead of the model predictions. Assume that any of these metrics is given by a sequence $z = \{z_1, \dots, z_T\}$ obtained from $T$ intermediate models. Then we can capture the uniformity of $z$ via a (weighted) variance score $s_{\mathsf{var}} = \sum_t w_t (z_t - \mu)^2$ for mean $\mu = \frac{1}{T} \sum_t z_t$ and an increasing weighting sequence $w = \{w_1, \dots, w_T\}$.

In order to show the effectiveness of the variance score $s_{\mathsf{var}}$ for continuous metrics, we provide a simple bound on the variance of confidence $\max_{y \in \mathcal{Y}} f_t(\boldsymbol{x})$ in the final checkpoints of the training. Assuming that the model has converged to a local minima with a low learning rate, we can assume that the distribution of model weights can be approximated by a Gaussian distribution.

We consider a linear regression problem where the inputs are linearly separable.

**Lemma 2.** *Assume that we have some Gaussian prior on the model parameters in the logistic regression setting across $m$ final checkpoints. More specifically, given $T$ total checkpoints of model parameters $\{\boldsymbol{w_1}, \boldsymbol{w_2}, \dots, \boldsymbol{w_T}\}$ we have $p(W = \boldsymbol{w_t}) = \mathcal{N}(\boldsymbol{w_0} \mid \boldsymbol{\mu}, s\boldsymbol{I})$ for $t \in \{T - m + 1, \dots, T\}$ and we assume that final checkpoints of the model are sampled from this distribution. We show that the variance of model confidence $\max_{y \in \{-1,1\}} p(y \mid \boldsymbol{x_i}, \boldsymbol{w_t})$ for a datapoint $(\boldsymbol{x_i}, y_i)$ can be upper bounded by a factor of probability of correctly classifying this example by the optimal weights.*

*Proof.* We first compute the variance of model predictions $p(y_i \mid \boldsymbol{x_i}, W)$ for a given datapoint $(\boldsymbol{x_i}, y_i)$. Following previous work (Schein & Ungar, 2007; Chang et al., 2017), the variance of predictions over these checkpoints can be estimated as follows:

Taking two terms in Taylor expansion for model predictions we have $p(y_i \mid \boldsymbol{x_i}, W) \simeq p(y_i \mid \boldsymbol{x_i}, \boldsymbol{w}) + g_i(\boldsymbol{w})^\top (W - \boldsymbol{w})$ where $W$ and $\boldsymbol{w}$ are current and the expected estimate of the parameters and $g_i(\boldsymbol{w}) = p(y_i \mid \boldsymbol{x_i}, \boldsymbol{w})(1 - p(y_i \mid \boldsymbol{x_i}, \boldsymbol{w}))\boldsymbol{x_i}$ is the gradient vector. Now we can write the variance with respect to the model prior as:

$$\mathbb{V}\left(p(y_i \mid \boldsymbol{x_i}, W)\right) \simeq \mathbb{V}\left(g_i(\boldsymbol{w})^\top (W - \boldsymbol{w})\right) = g_i(\boldsymbol{w})^\top F^{-1} g_i(\boldsymbol{w})$$

where $F$ is the variance of posterior distribution $p(W \mid X, Y) \sim \mathcal{N}(W \mid \boldsymbol{w}, F^{-1})$. This suggests that the variance of probability of correctly classifying $\boldsymbol{x}_i$ is proportional to $p(y_i \mid \boldsymbol{x}_i, \boldsymbol{w})^2(1 - p(y_i \mid \boldsymbol{x}_i, \boldsymbol{w}))^2$. Now we can bound the variance of maximum class probability or confidence as below:

$$\mathbb{V}\left(\max_{y \in \{-1,1\}} p(y \mid \boldsymbol{x}_i, W)\right) \leq \mathbb{V}\left(p(y_i \mid \boldsymbol{x}_i, W)\right) + \mathbb{V}\left(p(-y_i \mid \boldsymbol{x}_i, W)\right)$$

$$\approx 2p(y_i \mid \boldsymbol{x}_i, \boldsymbol{w})^2(1 - p(y_i \mid \boldsymbol{x}_i, \boldsymbol{w}))^2 \boldsymbol{x}_i^\top F^{-1} \boldsymbol{x}_i$$

$\square$

We showed that if the probability of correctly classifying an example given the final estimate of model parameters is close to one, the variance of model predictions following a Gaussian prior gets close to zero, we expect a similar behaviour for the variance of confidence under samples of this distribution.

## B  Extension of Empirical Evaluation

### B.1  Full Hyper-Parameters

We document full hyper-parameter settings for our method (**SPTD**) as well as all baseline approaches in Table 3.

Table 3: **Hyper-parameters used for all algorithms for classification.**

| Dataset | SC Algorithm | Hyper-Parameters |
|---|---|---|
| CIFAR-10 | Softmax Response (**SR**) | N/A |
| | Self-Adaptive Training (**SAT**) | $P = 100$ |
| | Deep Ensembles (**DE**) | $E = 10$ |
| | Selective Prediction Training Dynamics (**SPTD**) | $T = 1600, \ k = 2$ |
| CIFAR-100 | Softmax Response (**SR**) | N/A |
| | Self-Adaptive Training (**SAT**) | $P = 100$ |
| | Deep Ensembles (**DE**) | $E = 10$ |
| | Selective Prediction Training Dynamics (**SPTD**) | $T = 1600, \ k = 2$ |
| Food101 | Softmax Response (**SR**) | N/A |
| | Self-Adaptive Training (**SAT**) | $P = 100$ |
| | Deep Ensembles (**DE**) | $E = 10$ |
| | Selective Prediction Training Dynamics (**SPTD**) | $T = 2200, \ k = 3$ |
| StanfordCars | Softmax Response (**SR**) | N/A |
| | Self-Adaptive Training (**SAT**) | $P = 100$ |
| | Deep Ensembles (**DE**) | $E = 10$ |
| | Selective Prediction Training Dynamics (**SPTD**) | $T = 800, \ k = 5$ |

### B.2  Additional Selective Prediction Results

#### B.2.1  Extended Synthetic Experiments

We extend the experiment from Figure 2 to all tested SC methods in Figure 11. We also provide an extended result using Bayesian Linear Regression in Figure 12.

#### B.2.2  CIFAR-100 Results With ResNet-50

We further provide a full set of results using the larger ResNet-50 architecture on CIFAR-100 in Figure 4.

Table 4: **Selective accuracy achieved across coverage levels for CIFAR-100 with ResNet-50**.

| Cov. | SR | SAT+ER+SR | DE | SPTD | DE+SPTD |
|------|-----|-----------|-----|------|---------|
| 100 | **77.0 (±0.0)** | **77.0 (±0.0)** | **77.0 (±0.0)** | **77.0 (±0.0)** | **77.0 (±0.0)** |
| 90 | 79.2 (± 0.1) | 79.9 (± 0.1) | 81.2 (± 0.0) | 81.4 (± 0.1) | **82.1 (± 0.1)** |
| 80 | 83.1 (± 0.0) | 83.9 (± 0.0) | 85.7 (± 0.1) | 85.6 (± 0.1) | **86.0 (± 0.2)** |
| 70 | 87.4 (± 0.1) | 88.2 (± 0.1) | 89.6 (± 0.1) | **89.7 (± 0.0)** | 89.8 (± 0.1) |
| 60 | 90.5 (± 0.0) | 90.8 (± 0.2) | **90.7 (± 0.2)** | 90.6 (± 0.0) | 90.9 (± 0.1) |
| 50 | 93.4 (± 0.1) | 93.8 (± 0.0) | 95.3 (± 0.0) | 95.1 (± 0.0) | **95.4 (± 0.0)** |
| 40 | 95.4 (± 0.0) | 95.5 (± 0.1) | 97.1 (± 0.1) | **97.2 (± 0.1)** | 97.2 (± 0.0) |
| 30 | 97.4 (± 0.2) | 97.7 (± 0.0) | **98.6 (± 0.1)** | 98.6 (± 0.1) | 98.7 (± 0.0) |
| 20 | 97.9 (± 0.1) | 98.4 (± 0.1) | 99.0 (± 0.0) | 99.2 (± 0.1) | 99.2 (± 0.1) |
| 10 | 98.1 (± 0.0) | 98.8 (± 0.1) | 99.2 (± 0.1) | **99.4 (± 0.1)** | 99.6 (± 0.1) |

### B.2.3 Applying SPTD on Top of SAT

Our main set of results suggest that applying **SPTD** on top of **DE** further improves performance. The same effect holds when applying **SPTD** on top of non-ensemble-based methods such as **SAT**. We document this result in Figure 13.

### B.2.4 Ablation on $k$

We provide a comprehensive ablation on the weighting parameter $k$ in Figures 5 and 14.

### B.2.5 Comparison With Logit-Variance Approaches

We showcase the effectiveness of **SPTD** against **LOGITVAR** (Swayamdipta et al., 2020), an approach that also computes predictions of intermediate models but instead computes the variance of the correct prediction. We adapt this method to our selective prediction approach (for which true labels are not available) by computing the variance over the maximum predicted logit instead of the true logit. In Figure 15, we see that the weighting of intermediate checkpoints introduced by **SPTD** leads to stronger performance over the **LOGITVAR** baseline approach.

### B.2.6 Estimating $\tau$ on Validation VS Test Data

Consistent with prior works (Geifman & El-Yaniv, 2017; Liu et al., 2019; Huang et al., 2020; Feng et al., 2023), we estimate $\tau$ directly on the test set. However, a realistically deployable approach has to compute thresholds based on a validation set for which labels are available. In the case of selective classification, the training, validation, and test sets follow the i.i.d. assumption, which means that an approach that sets the threshold based on a validation set should work performantly on a test set, too. Under consistent distributional assumptions, estimating thresholds on a validation set functions as an unbiased estimator of accuracy/coverage tradeoffs on the test set. By the same virtue, setting thresholds directly on the test set and observing the SC performance on that test set should be indicative for additional test samples beyond the actual provided test set. It is important to remark that the validation set should only be used for setting the thresholds and not for model selection / early stopping which would indeed cause a potential divergence between SC performance on the validation and test sets. Further note that violations of the i.i.d assumption can lead to degraded performance due to mismatches in attainable coverage as explored in Bar-Shalom et al. (2023).

To confirm this intuition, we present an experiment in Figure 16 and Figure 17 where we select 50% of the samples from the test set as our validation set (and maintain the other 50% of samples as our new test set). We first generate 5 distinct such validation-test splits, set the thresholds for $\tau$ based on the validation set, and then evaluate selective classification performance on the test set by using the thresholds derived from

the validation set. We compare these results with our main approach which sets the thresholds based on the test set directly (ignoring the validation set). We provide an additional experiment where we partition the validation set from the training set in Figure 18. We see that the results are statistically indistinguishable from each other, confirming that this evaluation practice is valid for the selective classification setup we consider.

### B.2.7 Comparing $s_{\text{MAX}}$ and $s_{\text{SUM}}$

As per our theoretical framework and intuition provided in Section 3, the sum score $s_{\text{SUM}}$ should offer the most competitive selective classification performance. We confirm this finding in Figure 19 where we plot the accuracy/coverage curves across all datasets for both $s_{\text{MAX}}$ and $s_{\text{SUM}}$. Overall, we find that the sum score $s_{\text{SUM}}$ consistently outperforms the more noisy maximum score $s_{\text{MAX}}$.

### B.3 The Importance of Accuracy Alignment

Our results in Table 1 rely on accuracy alignment: We explicitly make sure to compare all methods on an equal footing by disentangling selective prediction performance from gains in overall utility. This is done by early stopping model training when the accuracy of the worst performing model is reached.

We expand on the important point that many previous approaches conflate both (i) generalization performance and (ii) selective prediction performance into a single score: the area under the accuracy/coverage curve. This metric can be maximized either by improving generalization performance (choosing different architectures or model classes) or by actually improving the ranking of points for selective prediction (accepting correct points first and incorrect ones last). As raised by a variety of recent works Geifman et al. (2018); Rabanser et al. (2023); Cattelan & Silva (2023), it is impossible and problematic to truly assess whether a method performs better at selective prediction (i.e., determining the correct acceptance ordering) without normalizing for this inherent difference yielded as a side effect by various SC methods. In other words, an SC method with lower base accuracy (smaller correct set) can still outperform another SC method with higher accuracy (larger correct set) in terms of the selective acceptance ordering (an example of which is given in Table 3 of Liu et al. (2019)). Accuracy normalization allows us to eliminate these confounding effects between full-coverage utility and selective prediction performance by identifying which models are better at ranking correct points first and incorrect ones last. This is of particular importance when comparing selective prediction methods which change the training pipeline in different ways, as is done in the methods presented in Table 1.

However, when just comparing **SPTD** to one other method, we do not need to worry about accuracy normalization. Showcasing this, we run **SPTD** on top of an unnormalized **SAT+ER+SR** run and provide these experiments in Figure 13. We see that the application of **SPTD** on top of **SAT+ER+SR** allows us to further boost performance (similar to the results where we apply **SPTD** on top of **DE** in Table 1). So to conclude, experimentally, when using the best model, we see that **SPTD** still performs better at selective prediction than the relevant baseline for that training pipeline. We wish to reiterate that this issue of accuracy normalization highlights another merit of **SPTD**, which is that it can easily be applied on top of any training pipeline (including those that lead to the best model) and allows easy comparison to the selective classification method that training pipeline was intended to be deployed with.

### B.4 Evaluation using other performance metrics

We further provide results of summary performance metrics across datasets in Table 5:

- The area under the accuracy-coverage curve (`AUACC`) as discussed in Geifman et al. (2018).

- The area under the receiver operating characteristic (`AUROC`) as suggested by Galil et al. (2023).

- The accuracy normalized selective classification score (`ANSC`) from Geifman et al. (2018) and Rabanser et al. (2023).

Table 5: **Evaluation of SC approaches using various evaluation metrics**.

| Dataset | Method | $1 - $ AUACC | ANSC | AUROC |
|---|---|---|---|---|
| CIFAR10 | SR | $0.053 \pm 0.002$ | $0.007 \pm 0.000$ | $0.918 \pm 0.002$ |
| | SPTD | $0.048 \pm 0.001$ | $0.004 \pm 0.000$ | $0.938 \pm 0.002$ |
| | DE | $0.046 \pm 0.002$ | $0.004 \pm 0.000$ | $0.939 \pm 0.003$ |
| | SAT | $0.054 \pm 0.002$ | $0.006 \pm 0.000$ | $0.924 \pm 0.005$ |
| | DG | $0.054 \pm 0.001$ | $0.006 \pm 0.000$ | $0.922 \pm 0.005$ |
| CIFAR100 | SR | $0.181 \pm 0.001$ | $0.041 \pm 0.001$ | $0.865 \pm 0.003$ |
| | SPTD | $0.174 \pm 0.002$ | $0.037 \pm 0.000$ | $0.872 \pm 0.002$ |
| | DE | $0.159 \pm 0.001$ | $0.030 \pm 0.001$ | $0.880 \pm 0.003$ |
| | SAT | $0.180 \pm 0.001$ | $0.041 \pm 0.001$ | $0.866 \pm 0.003$ |
| | DG | $0.182 \pm 0.001$ | $0.041 \pm 0.001$ | $0.867 \pm 0.002$ |
| GTSRB | SR | $0.020 \pm 0.002$ | $0.001 \pm 0.000$ | $0.986 \pm 0.003$ |
| | SPTD | $0.019 \pm 0.002$ | $0.001 \pm 0.000$ | $0.986 \pm 0.005$ |
| | DE | $0.015 \pm 0.001$ | $0.001 \pm 0.000$ | $0.986 \pm 0.002$ |
| | SAT | $0.027 \pm 0.001$ | $0.001 \pm 0.000$ | $0.984 \pm 0.002$ |
| | DG | $0.019 \pm 0.003$ | $0.001 \pm 0.000$ | $0.986 \pm 0.002$ |
| SVHN | SR | $0.027 \pm 0.000$ | $0.006 \pm 0.001$ | $0.895 \pm 0.004$ |
| | SPTD | $0.025 \pm 0.003$ | $0.003 \pm 0.001$ | $0.932 \pm 0.005$ |
| | DE | $0.021 \pm 0.001$ | $0.005 \pm 0.000$ | $0.912 \pm 0.003$ |
| | SAT | $0.028 \pm 0.001$ | $0.006 \pm 0.000$ | $0.895 \pm 0.002$ |
| | DG | $0.026 \pm 0.001$ | $0.007 \pm 0.000$ | $0.896 \pm 0.006$ |

## B.5 Evaluation of more competing approaches

We further compare our method with two more contemporary selective prediction approaches:

- **AUCOC** (Sangalli et al., 2024): This work uses a custom cost function for multi-class classification that accounts for the trade-off between a neural network's accuracy and the amount of data that requires manual inspection from a domain expert.

- **CCL**-**SC** (Wu et al., 2024): This work proposes optimizing feature layers to reduce intra-class variance via contrastive learning.

Across both methods, we find that they do not outperform ensemble-based methods like **DE** and hence also do not outperform **SPTD**. See Table 6 for detailed results.

Table 6: **Selective accuracy achieved across coverage levels for AUCOC and CCL-SC**. Similar as Table 1. Neither **AUCOC** nor **CCL-SC** is able to outperform **DE** or **SPTD**. **Bold** numbers are best results at a given coverage level across all methods and underlined numbers are best results for methods relying on a single training run only.

|  | Coverage | AUCOC | CCL-SC | SPTD | DE |
|---|---|---|---|---|---|
| *CIFAR-10* | 100 | **92.9** ($\pm$0.0) | **92.9** ($\pm$0.0) | **92.9** ($\pm$0.0) | **92.9** ($\pm$0.0) |
|  | 90 | 96.0 ($\pm$0.1) | 95.9 ($\pm$0.2) | 96.5 ($\pm$0.0) | **96.8** ($\pm$0.1) |
|  | 80 | 98.1 ($\pm$0.2) | 98.0 ($\pm$0.3) | 98.4 ($\pm$0.1) | **98.7** ($\pm$0.0) |
|  | 70 | 99.0 ($\pm$0.3) | 98.5 ($\pm$0.2) | 99.2 ($\pm$0.1) | **99.4** ($\pm$0.1) |
|  | 60 | 99.3 ($\pm$0.1) | 99.1 ($\pm$0.2) | **99.6** ($\pm$0.2) | **99.6** ($\pm$0.1) |
|  | 50 | 99.4 ($\pm$0.2) | 99.0 ($\pm$0.3) | **99.8** ($\pm$0.0) | 99.7 ($\pm$0.0) |
|  | 40 | 99.5 ($\pm$0.1) | 99.4 ($\pm$0.2) | **99.8** ($\pm$0.1) | **99.8** ($\pm$0.0) |
|  | 30 | 99.5 ($\pm$0.2) | 99.2 ($\pm$0.3) | **99.8** ($\pm$0.1) | **99.8** ($\pm$0.0) |
|  | 20 | 99.6 ($\pm$0.1) | 99.4 ($\pm$0.2) | **100.0** ($\pm$0.0) | 99.8 ($\pm$0.0) |
|  | 10 | 99.7 ($\pm$0.0) | 99.4 ($\pm$0.1) | **100.0** ($\pm$0.0) | 99.8 ($\pm$0.0) |
| *CIFAR-100* | 100 | **75.1** ($\pm$0.0) | **75.1** ($\pm$0.0) | **75.1** ($\pm$0.0) | **75.1** ($\pm$0.0) |
|  | 90 | 78.7 ($\pm$0.2) | 76.5 ($\pm$0.3) | **80.4** ($\pm$0.0) | 80.2 ($\pm$0.0) |
|  | 80 | 83.2 ($\pm$0.1) | 82.2 ($\pm$0.2) | 84.6 ($\pm$0.1) | **84.7** ($\pm$0.1) |
|  | 70 | 87.4 ($\pm$0.1) | 86.1 ($\pm$0.2) | **88.7** ($\pm$0.0) | 88.6 ($\pm$0.1) |
|  | 60 | 89.8 ($\pm$0.2) | 88.6 ($\pm$0.3) | 90.1 ($\pm$0.0) | **90.2** ($\pm$0.2) |
|  | 50 | 93.3 ($\pm$0.1) | 92.1 ($\pm$0.2) | 94.6 ($\pm$0.0) | **94.8** ($\pm$0.0) |
|  | 40 | 95.9 ($\pm$0.2) | 95.2 ($\pm$0.3) | **96.9** ($\pm$0.1) | 96.8 ($\pm$0.1) |
|  | 30 | 98.2 ($\pm$0.1) | 96.6 ($\pm$0.2) | **98.4** ($\pm$0.1) | 98.4 ($\pm$0.1) |
|  | 20 | 98.6 ($\pm$0.2) | 98.4 ($\pm$0.3) | 98.8 ($\pm$0.2) | **99.0** ($\pm$0.0) |
|  | 10 | 98.8 ($\pm$0.1) | 98.7 ($\pm$0.2) | **99.4** ($\pm$0.1) | 99.2 ($\pm$0.1) |

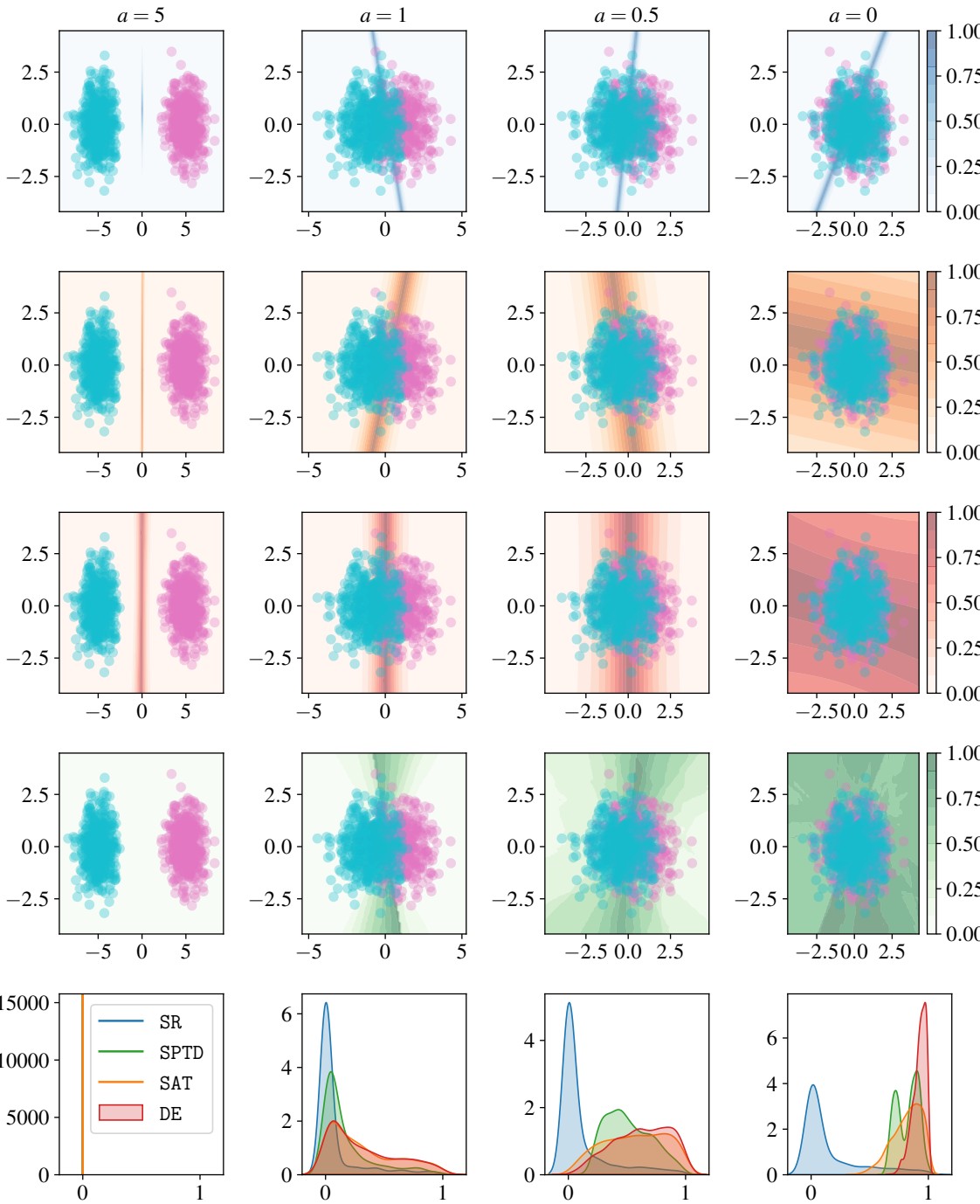

Figure 11: **Extended Gaussian experiment.** The first row corresponds to the anomaly scoring result of **SR**, the second to the result of **SAT**, the third to the result of **DE**, and the fourth to the result of **SPTD**. The bottom row shows the score distribution for each method over the data points. We see that all methods reliably improve over the **SR** baseline. At the same time, we notice that **SAT** and **DE** still assign higher confidence away from the data due to limited use of decision boundary oscillations. **SPTD** addresses this limitation and assigns more uniform uncertainty over the full data space.

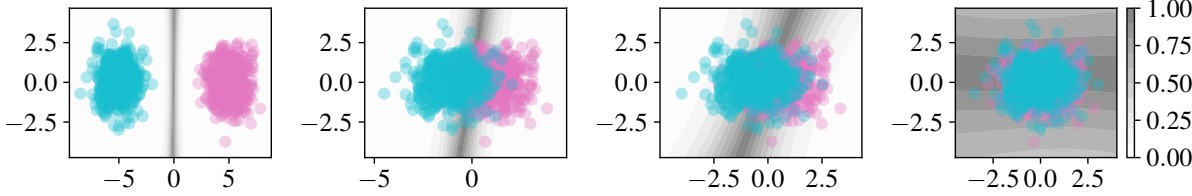

Figure 12: **Bayesian linear regression experiment on Gaussian data.** Results comparable to **DE**.

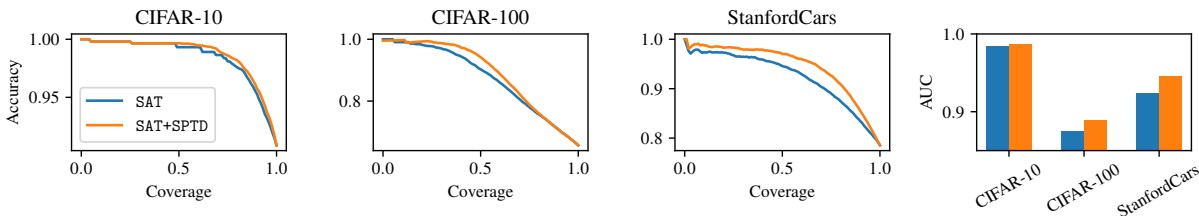

Figure 13: **Applying SPTD on top of SAT.** Similar as with **DE**, we observe that the application of **SPTD** improves performance.

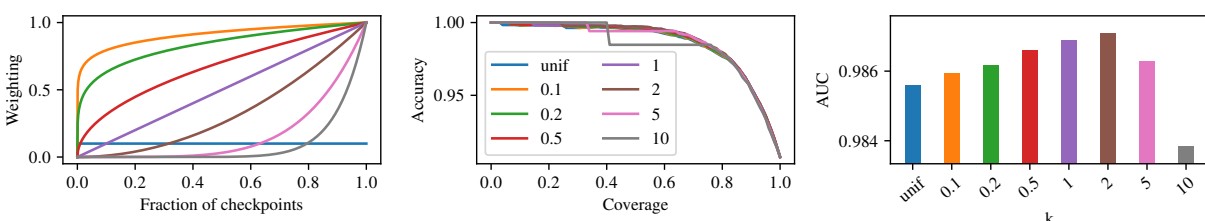

Figure 14: **Extended ablation results on $k$ on CIFAR-10.** We now also consider $k \in (0, 1]$ as well as a uniform weighting assigning the same weight to all checkpoints. We confirm that a convex weighting yields best performance.

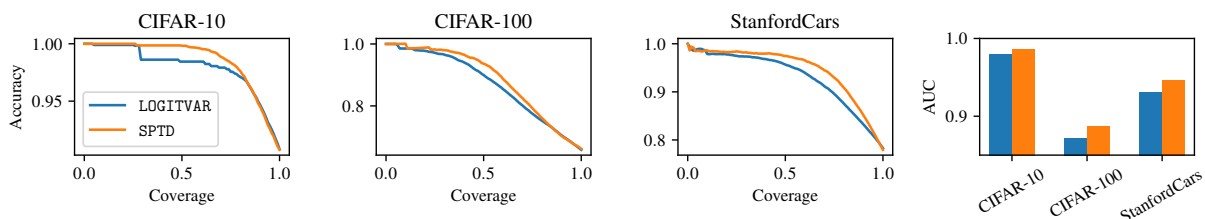

Figure 15: **Comparison of LOGITVAR vs SPTD.** We observe that **SPTD**, which incorporates weighting of intermediate checkpoints using $v_t$, outperforms **LOGITVAR**.

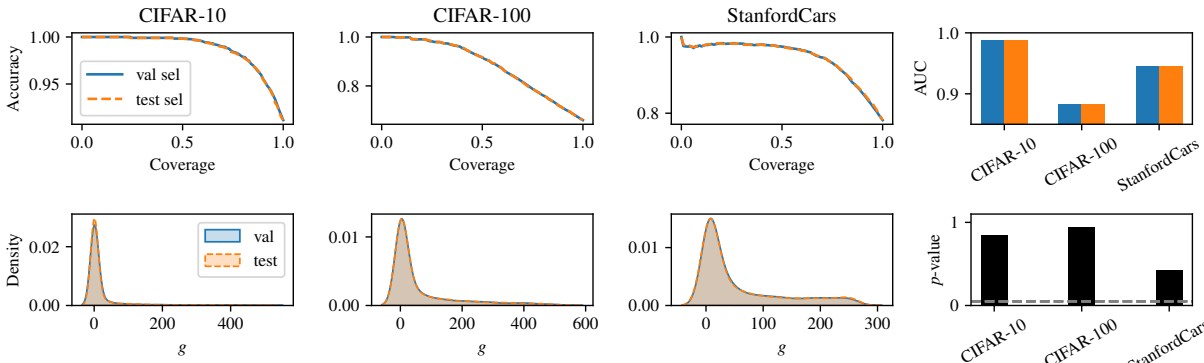

Figure 16: **SPTD accuracy/coverage trade-offs and score distributions on test data obtained by computing $\tau$ on a validation set or directly on the test set.** The first row shows the obtained accuracy/coverage trade-offs with the respective AUC scores. In the second row, we show the score distribution for both the picked validation and test sets, along with $p$-values from a KS-test to determine the statistical closeness of the distributions. Overall, we observe that both methods are statistically indistinguishable from each other.

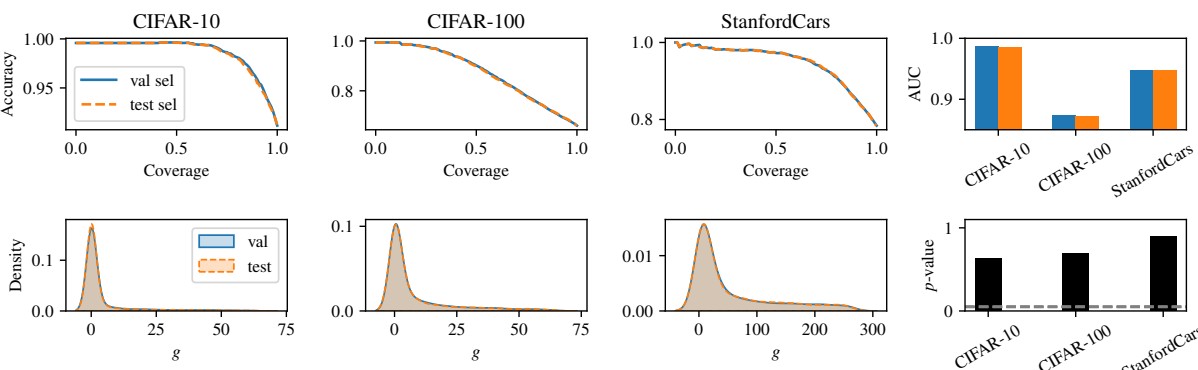

Figure 17: **SAT accuracy/coverage trade-offs and score distributions on test data obtained by computing $\tau$ on a validation set or directly on the test set.** Same as Figure 16 but with **SAT**.

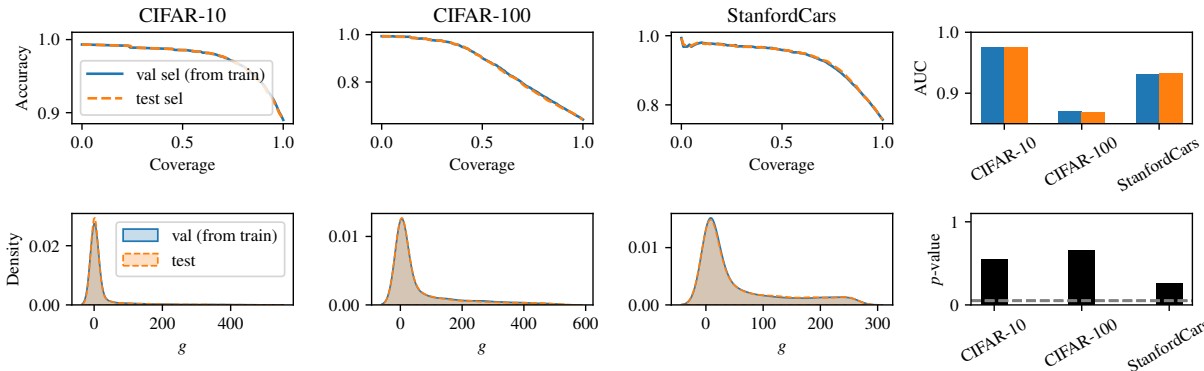

Figure 18: **SPTD accuracy/coverage trade-offs and score distributions on test data obtained by computing $\tau$ on a validation set or directly on the test set.** Same as Figure 16 but with the validation set is taken from the original training set.

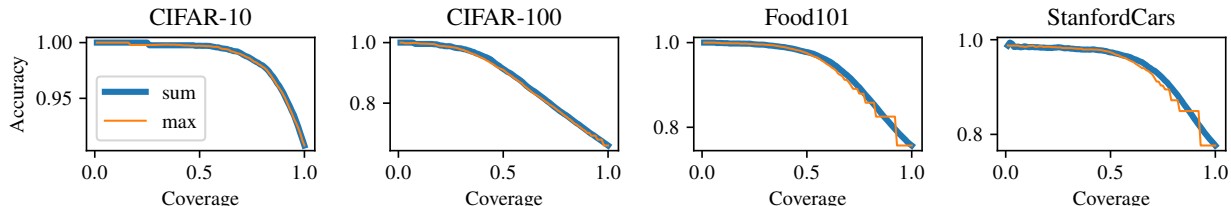

Figure 19: **Comparing $s_{\text{MAX}}$ and $s_{\text{SUM}}$ performance**. It is clear that $s_{\text{SUM}}$ effectively denoises $s_{\text{MAX}}$.

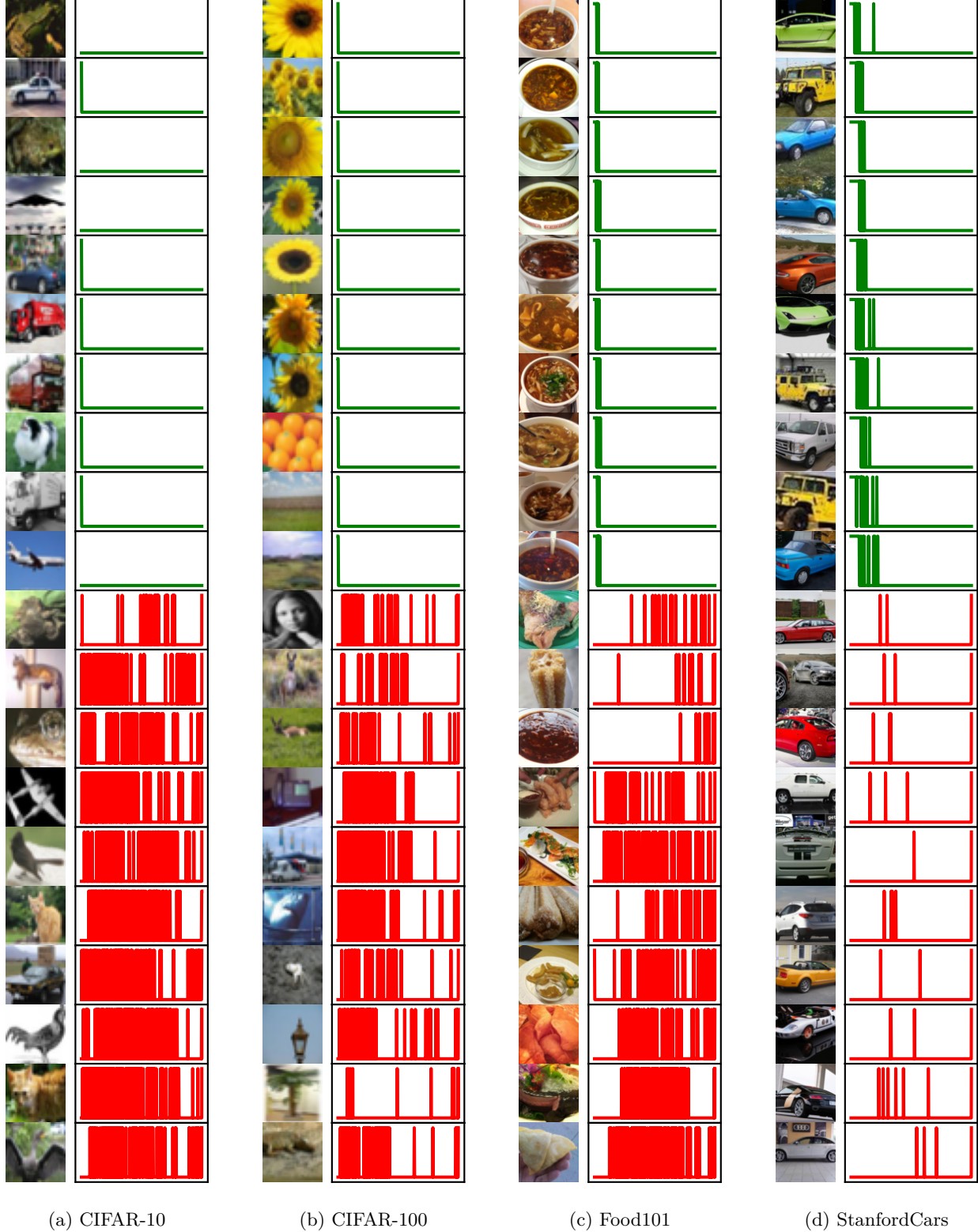

(a) CIFAR-10  (b) CIFAR-100  (c) Food101  (d) StanfordCars

Figure 20: **Additional individual examples across datasets.**

