# OpenReview forum: "Selective Prediction via Training Dynamics"
_TMLR — Accepted by TMLR_

### Review · Reviewer_5C8G · 2024-09-26

**Summary Of Contributions:**

The paper proposes a selective prediction method based on training dynamics. The method saves multiple checkpoints during training. Each test sample is passed through. Then, a score for selective classification is computed based on the consistency between the predictions at each checkpoint and the final prediction. The predictions for the samples that have high uncertainty are rejected.

The paper proposes different score functions for classification, regression, and time series prediction problems and performs extensive experiments to compare the proposed method with the existing methods on multiple benchmark datasets. The results demonstrate that the method improves the performance of the existing methods. Additionally, ablation studies are performed to show the effect of crucial hyperparameters of the method on the performance.

**Audience:**

Yes

**Broader Impact Concerns:**

I don't have any broader impact concern.

**Claims And Evidence:**

Yes

**Requested Changes:**

- Perform comparisons with more recent works such as Sangalli et al. and Wu et al.
- Perform comparisons with the works that are close to the proposed method, e.g. Agarwal et al.
- Demonstrate the performance of the method when the network is trained with different losses tailored for selective classification to demonstrate that the method can be complementary to the existing methods.

**Strengths And Weaknesses:**

Strengths:
- The idea of using training dynamics for selective classification is interesting.
- The experiments are extensive and demonstrate the advantage of the proposed method.
- Sensitivity of the proposed method to it's hyperparameters is evaluated in ablation studies.

Weakness:
- Although the paper presents extensive experiments, it misses comparison with some recent works such as [1, 2].

[1] Sangalli et al. Expert load matters: operating networks at high accuracy and low manual effort, Neurips 2023.
[2] Wu et al. Confidence-aware Contrastive Learning for Selective Classification, ICML 2024

- The paper mentions Agarwal et al. (2020) as work that is closest to the proposed method; however, the proposed method is not compared to it.

- There are many "loss medications" based methods in the literature as also mentioned in the paper. And the proposed method can be complementary to those and one can get checkpoints during training with different losses. I think it would be an interesting experiment to show the performance of the method when training with various losses.

---

> ### Author Response · Authors · 2024-11-18
> **Review Response**
>
> We thank the reviewer for their constructive feedback and address concerns below:
>
> > Although the paper presents extensive experiments, it misses comparison with some recent works such as [1, 2].
>
> As suggested by the reviewer, we have added additional comparisons for both methods for CIFAR-10 and CIFAR-100. In summary, we find that neither of the approaches outperform SPTD (or Deep Ensembles). We have also updated the paper with a new Section B.5 and a results table (Table 6).
>
> CIFAR-10:
> | Coverage | AUCOC            | CCL-SC           | SPTD             | DE               |
> |----------|------------------|------------------|------------------|------------------|
> | 100      | **92.9** (±0.0)  | **92.9** (±0.0)  | **92.9** (±0.0)  | **92.9** (±0.0)  |
> | 90       | 96.0 (±0.1)      | 95.9 (±0.2)      | _96.5_ (±0.0)    | **96.8** (±0.1)  |
> | 80       | 98.1 (±0.2)      | 98.0 (±0.3)      | _98.4_ (±0.1)    | **98.7** (±0.0)  |
> | 70       | 99.0 (±0.3)      | 98.5 (±0.2)      | _99.2_ (±0.1)    | **99.4** (±0.1)  |
> | 60       | 99.3 (±0.1)      | 99.1 (±0.2)      | **_99.6_** (±0.2)| **99.6** (±0.1)  |
> | 50       | 99.4 (±0.2)      | 99.0 (±0.3)      | **_99.8_** (±0.0)| 99.7 (±0.0)      |
> | 40       | 99.5 (±0.1)      | 99.4 (±0.2)      | **_99.8_** (±0.1)| **99.8** (±0.0)  |
> | 30       | 99.5 (±0.2)      | 99.2 (±0.3)      | **_99.8_** (±0.1)| **99.8** (±0.0)  |
> | 20       | 99.6 (±0.1)      | 99.4 (±0.2)      | **_100.0_** (±0.0)| 99.8 (±0.0)     |
> | 10       | 99.7 (±0.0)      | 99.4 (±0.1)      | **_100.0_** (±0.0)| 99.8 (±0.0)     |
>
> CIFAR-100:
> | Coverage | AUCOC            | CCL-SC           | SPTD             | DE               |
> |----------|------------------|------------------|------------------|------------------|
> | 100      | **_75.1_** (±0.0)| **_75.1_** (±0.0)| **_75.1_** (±0.0)| **_75.1_** (±0.0)|
> | 90       | 78.7 (±0.2)      | 76.5 (±0.3)      | **_80.4_** (±0.0)| 80.2 (±0.0)      |
> | 80       | 83.2 (±0.1)      | 82.2 (±0.2)      | **_84.6_** (±0.1)| **84.7** (±0.1)  |
> | 70       | 87.4 (±0.1)      | 86.1 (±0.2)      | **_88.7_** (±0.0)| 88.6 (±0.1)      |
> | 60       | 89.8 (±0.2)      | 88.6 (±0.3)      | _90.1_ (±0.0)    | **90.2** (±0.2)  |
> | 50       | 93.3 (±0.1)      | 92.1 (±0.2)      | _94.6_ (±0.0)    | **94.8** (±0.0)  |
> | 40       | 95.9 (±0.2)      | 95.2 (±0.3)      | **_96.9_** (±0.1)| 96.8 (±0.1)      |
> | 30       | 98.2 (±0.1)      | 96.6 (±0.2)      | **_98.4_** (±0.1)| 98.4 (±0.1)      |
> | 20       | 98.6 (±0.2)      | 98.4 (±0.3)      | _98.8_ (±0.2)    | **99.0** (±0.0)  |
> | 10       | 98.8 (±0.1)      | 98.7 (±0.2)      | **_99.4_** (±0.1)| 99.2 (±0.1)      |
>
> > The paper mentions Agarwal et al. (2020) as work that is closest to the proposed method; however, the proposed method is not compared to it.
>
> The reason our work does not directly compare with many training dynamics based approaches such as Swayamdipta et al. (2020) (variance of logits) and Agarwal et al. (2020) (variance of gradients) in the main paper body is that both approaches rely on true label information. As such, these techniques have been used to analyze training behavior but have not been applied to selective prediction where true label information is not available at test time. However, both approaches can be adapted to work as selective prediction approaches by replacing their respective variance of logits/gradients calculations w.r.t the predicted label. The current paper draft already includes this experiment in Appendix B.2.5 where we show that SPTD outperforms this adapted version of variance of logits.
>
> > There are many "loss medications" based methods in the literature as also mentioned in the paper. And the proposed method can be complementary to those [...].
>
> The reviewer is right that our approach can be flexibly composed on top of existing training pipelines that use different losses (or optimization procedures) to improve selective prediction performance. To showcase this flexibility, the current paper draft already includes two performant selective prediction approaches:
> - DE+SPTD (Section 4, Table 1): Here, we apply SPTD on top of every single ensemble run of a Deep Ensemble. As discussed in the main body of the paper, this yields state of the art selective prediction performance.
> - SAT+SPTD (Appendix B.2.3, Figure 13): Here, we apply SPTD on top of a model trained with SAT, an approach that modifies the loss with a regularizer. We observe that applying SPTD on top of SAT also improves selective prediction performance.
>
> In summary, we find that applying SPTD on top of other selective prediction approaches further boost performance. We have added the following pointer to the SAT experiments on page 13:
> > Further evidence towards this flexibility is provided in Appendix B.2.3 where we show that applying SPTD on top of SAT also improves selective prediction performance.
>
> ---
>
> We are happy to further engage with the reviewer in case there are any remaining concerns.

---

### Review · Reviewer_Qa8X · 2024-10-15

**Summary Of Contributions:**

The paper introduces a novel method for selective prediction, termed SPTD, which is based on an ensemble approach that incorporates a gating function defined as $g(x) = \sum_{1 \leq t \leq T} v_t a_t(x)$. SPTD demonstrates strong performance across various tasks, including regression, time series analysis, and classification.

**Audience:**

Yes

**Claims And Evidence:**

Yes

**Requested Changes:**

- A discussion on the technical differences between SPTD and Deep Ensemble methods.
- A discussion on the possible feasibility of scaling the performance of SPTD to models such as ViT, which possess significantly more parameters.

**Strengths And Weaknesses:**

**Strengths:**
- The method extends the application of selective predication to regression and time-series prediction.
- The ablations are well-justified. The paper is well-written.


**Weaknesses:**
- This paper does not justify the selection of checkpoints as a reasonable approach for measuring uncertainty in the ensemble method of sub-models.
- In SPTD, saving checkpoints exceeding 1k for uncertainty measurement is not very feasible in practice, especially when the model has a large number of parameters.

---

> ### Author Response · Authors · 2024-11-18
> **Review Response**
>
> We thank the reviewer for their constructive feedback and address concerns below:
>
> > This paper does not justify the selection of checkpoints as a reasonable approach for measuring uncertainty in the ensemble method of sub-models.
>
> We have opted for sampling checkpoints at regular, linearly spaced intervals. This allows us to make sure that all parts of the training process are well represented in terms of the training dynamics. We have also tried other sampling strategies such as (i) sampling checkpoints according to a Beta distribution over the training process with varying $\alpha$ and $\beta$; and (ii) only considering the last $m \in [10,25,50]$ checkpoints obtained at the end of training. We did not find evidence for these more sophisticated sampling strategies to consistently improve our selective prediction performance. Although we do think that more carefully crafted checkpoint sampling strategies can be derived, their applicability and performance might be highly problem/dataset specific. Hence, we are confident that the regular sampling we are currently employing is a good default choice and additional tuning of this sampling can be performed as part of future work.
>
> > In SPTD, saving checkpoints exceeding 1k for uncertainty measurement is not very feasible in practice, especially when the model has a large number of parameters.
>
> The reviewer is right that we do indeed initially approximate the training dynamics on a very granular scale, leading to more than 1k checkpoints per dataset. As forward-propagating through > 1k checkpoints at test time is evidently prohibitive, we show in the Checkpoint Selection Strategy paragraph on page 13 that subsampling 50, 25, or even 10 checkpoints still leads to strong selective classification. Especially for high targeted coverage (>50%), the performance across all checkpointing resolutions is indistinguishable from each other (Figure 6). This insight allows us to reduce the computational overhead of our method to a similar cost as Deep Ensembles at test time while having a considerably less expensive training stage. Practical implementations of our method can directly store checkpoints at a more coarse-grained resolution.
>
> > A discussion on the technical differences between SPTD and Deep Ensemble methods.
>
> As described in Section 2.1, Deep Ensembles are composed of a collection of multiple models trained with different hyper-parameters until convergence. In contrast, our SPTD method, as described in the introduction and throughout Section 1 and 3, can be interpreted as an ensemble of models over a single training run. As such, these methods are distinct and composable; this is discussed in Section 4 (Accuracy/Coverage Trade-off paragraph) and Table 1.
>
> > A discussion on the possible feasibility of scaling the performance of SPTD to models such as ViT, which possess significantly more parameters.
>
> We have previously experimented with VGG architectures and have found our results to be consistent independent of the chosen architecture. We remark that ViT models have not yet shown to consistently provide better utility compared to ResNets on the datasets we consider [1] and hence opted against ViT experimentation in our work. To still showcase our method with larger model complexity, we show results for CIFAR-100 on a Resnet-50 architecture (which has about the same number of parameters as a ViT-small) in Table 4. We observe that the effectiveness of SPTD carries over to this larger architecture.
>
> **References**:
>
> [1] Zhu, Haoran, Boyuan Chen, and Carter Yang. "Understanding Why ViT Trains Badly on Small Datasets: An Intuitive Perspective." arXiv preprint arXiv:2302.03751 (2023).
>
> ---
>
> We are happy to further engage with the reviewer in case there are any remaining concerns.

---

### Review · Reviewer_BCan · 2024-11-05

**Summary Of Contributions:**

This article addresses the problem of selective prediction by monitoring prediction instability during training. Selective prediction aims to reject inputs that the model may misclassify, balancing input coverage with model utility. The proposed method, SPTD, leverages the dynamics of neural network training. Unlike existing methods that rely solely on the final model, SPTD utilizes intermediate models generated during iterative optimization processes like SGD. A prediction instability score is calculated to measure the reliability of predictions, with higher scores indicating greater uncertainty. This score informs a selection function that decides whether to predict a test point. SPTD is applicable to classification, regression, and time series forecasting problems. Experimental results demonstrate that SPTD achieves new state-of-the-art accuracy/coverage trade-offs across multiple datasets, significantly improving prediction accuracy while maintaining high coverage compared to existing methods.

**Audience:**

Yes

**Claims And Evidence:**

Yes

**Requested Changes:**

Please Check Weakness, by the way, this paper has some confused descriptions, such as "and disregard the rich set of statistics that are contained (or derivable) from this model sequence.", please revise the writing of the paper, Make it easier for readers to understand.

**Strengths And Weaknesses:**

**Strengths**

1. High Compatibility and Flexibility: Compatible with existing model architectures and training objectives, selective prediction methods without any additional training modifications. In addition, it is competent in SP tasks other than just classification: regression and time series prediction tasks. (I enjoy it! 😁)

2. Rich experimental verification and reasonable conclusions

**Weaknesses**

1. Checkpoint Selection: The fixed schedule for choosing checkpoints may lead to suboptimal model performance.

2. Hyperparameter Complexity: The method involves several hyperparameters, such as the number of checkpoints and weights for calculating the selection function, making it unclear which intermediate checkpoints are optimal for inference.

3. Resource Overhead: The method incurs additional storage and inference costs due to the need for multiple intermediate models

4. Lack of citations from papers in similar fields:

   [1] Assessing Generalization of SGD via Disagreement, ICLR 2022.

   [2] Agreement-on-the-Line Predicting the Performance of Neural Networks under Distribution Shift, Neurips2022.

   [3] Energy-based Automated Model Evaluation, ICLR 2024.

   [4] CAME: Contrastive Automated Model Evaluation, ICCV 2023.

---

> ### Author Response · Authors · 2024-11-18
> **Review Response**
>
> We thank the reviewer for their constructive feedback and address concerns below:
>
> > Checkpoint Selection: The fixed schedule for choosing checkpoints may lead to suboptimal model performance.
>
> We have tried other sampling strategies such as (i) sampling checkpoints according to a Beta distribution over the training process with varying $\alpha$ and $\beta$; and (ii) only considering the last $m \in [10,25,50]$ checkpoints obtained at the end of training. We did not find evidence for these more sophisticated sampling strategies to consistently improve our selective prediction performance. Although we do think that more carefully crafted checkpoint sampling strategies can be derived, their applicability and performance might be highly problem/dataset specific. Hence, we are confident that the regular sampling we are currently employing is a good default choice and additional tuning of this sampling can be performed as part of future work.
>
> > Hyperparameter Complexity: The method involves several hyperparameters, such as the number of checkpoints and weights for calculating the selection function, making it unclear which intermediate checkpoints are optimal for inference.
>
> See discussion on previous point where we found fixed checkpointing intervals to work best, and note that we found more checkpoints to always be better. Hence one only needs to decide how many checkpoints they can store and run inference on, and checkpoint regularly throughout training accordingly to obtain a good default performance (without hyperparameter tuning).
> We also want to alert the reviewer to an additional ablation documented in Figure 14 where we experiment with ablations on the weighting hyperparameter $k$ on CIFAR-10. Here, we consider both concave $k \in [0.1,1)$ and convex $k \in [1,10]$ as well as a uniform weighting assigning the same weight to all checkpoints. We confirm that a convex weighting yields best performance and also found that a fixed value of $k \in [2,5]$ yielded the most competitive results across all datasets. Hence, we believe $k$ is not a hyperparameter that needs to be significantly tuned, similar to checkpointing.
>
> > Resource Overhead: The method incurs additional storage and inference costs due to the need for multiple intermediate models
>
> Indeed, our method relies on a more costly inference stage as documented in Table 2. However, our method offers several other advantages over most existing methods: i) no changes required for the training stage (opening up the potential for retrofitting our method to existing models in the presence of checkpoints); ii) compositionality with existing selective prediction approaches; (iii) applicability beyond classification; and (iv) new SOTA SP performance.
>
> > Lack of citations from papers in similar fields.
>
> We thank the reviewer for alerting us to these papers. We have updated our manuscript to include these as relevant background.
>
>
> > by the way, this paper has some confused descriptions, such as "and disregard the rich set of statistics that are contained (or derivable) from this model sequence.", please revise the writing of the paper, Make it easier for readers to understand.
>
> We have checked the paper for readability and revised this sentence in particular.
>
> ---
>
> We are happy to further engage with the reviewer in case there are any remaining concerns.

---

> > ### Comment · Reviewer_BCan · 2024-12-15
> > **Response for Authors**
> >
> > Thanks the authors for the detailed response. I think my concerns and questions are well-addressed. With the revisions and new experiments added in the paper, it is a solid paper to me. My attitude towards this paper tends to be acceptable.

---

### Decision · Action_Editor_XZwp · 2024-12-31

**Recommendation:** Accept as is

**Comment:**

All reviewers commended the novelty of the proposed method, The paper is also well-written.  Two reviewers officially endorsed the acceptance. While the third reviewer (Reviewer BCan) did not provide the official recommendation, they commented in response to the authors' response that their concerns and questions were well-addressed and recommended its acceptance.

Based on the reviewers' recommendations and my reading of the paper, I recommend its acceptance.

**Audience:**

Selective prediction is an important topic in machine/deep learning.  The proposed method is easy to implement and practically useful. As such, this paper will surely be of interest to the TMLR audience.

**Claims And Evidence:**

The paper proposed an intriguing method for selective prediction that rejects data points that exhibit too much disagreement with the final prediction at the late stages of SGD training.  The claimed effectiveness is well supported by extensive experiments compared to SOTA.